# Empowering Active Learning for 3D Molecular Graphs with Geometric Graph Isomorphism

**Ronast Subedi**[*]
Florida State University
rs22ce@fsu.edu

**Lu Wei**[*]
Stony Brook University
lu.wei.1@stonybrook.edu

**Wenhan Gao**[*]
Stony Brook University
wenhan.gao@stonybrook.edu

**Shayok Chakraborty**[†]
Florida State University
shayok@cs.fsu.edu

**Yi Liu**[†]
Stony Brook University
yi.liu.4@stonybrook.edu

## Abstract

Molecular learning is pivotal in many real-world applications, such as drug discovery. Supervised learning requires heavy human annotation, which is particularly challenging for molecular data; *e.g.*, the commonly used density functional theory (DFT) is highly computationally expensive. Active learning (AL) automatically queries labels for the most informative samples, thereby remarkably alleviating the annotation hurdle. In this paper, we present a principled AL paradigm for molecular learning, where we treat molecules as 3D molecular graphs. Specifically, we propose a new diversity sampling method to eliminate mutual redundancy built on distributions of 3D geometries. We first propose a set of new 3D graph isometries for 3D graph isomorphism analysis. Our method is provably at least as expressive as the Geometric Weisfeiler-Lehman (GWL) test. The moments of the distributions of the associated geometries are then extracted for efficient diversity computing. To ensure our AL paradigm selects samples with maximal uncertainties, we carefully design a Bayesian geometric graph neural network to compute uncertainties specifically for 3D molecular graphs. We pose active sampling as a quadratic programming (QP) problem using the proposed components. Experimental results demonstrate the effectiveness of our AL paradigm, as well as the proposed diversity and uncertainty methods. The code is publicly available at https://github.com/sronast/al_3dgraph.

## 1 Introduction

Molecular representation learning is essential for various real-world applications, such as molecular design, drug discovery, material design, *etc.*. In recent studies, molecules have been formulated as 3D graphs, based on the evidence that 3D spatial information is crucial to determine the properties of molecules [Liu et al., 2019, Townshend et al., 2019, Axelrod and Gomez-Bombarelli, 2020]. Generally, in a 3D graph, atoms are represented as nodes, each associated with Cartesian coordinates in 3D space. A predefined cut-off distance can be used as a threshold to determine if there is an edge between two nodes in the 3D graph. With the advance of deep learning, 3D graph neural networks (GNNs) have been developed to learn from 3D molecular graph data [Thomas et al., 2018, Schütt et al., 2017, Satorras et al., 2021, Gasteiger et al., 2020c, Liu et al., 2021, 2022, Wang et al., 2022, Liao and Smidt, 2022, Zhou et al., 2022, Yan et al., 2022, Wang et al., 2023, Lin et al., 2023, Zhang et al., 2023]. These models are data-hungry and necessitate a large amount of annotated training data

---

[*]Equal contribution.
[†]Corresponding author.

38th Conference on Neural Information Processing Systems (NeurIPS 2024).

to attain good performance. However, annotation usually consumes excessive manpower, which is particularly challenging for molecules, *e.g.*, the commonly used density functional theory (DFT) for molecular energy computing [Hohenberg and Kohn, 1964] is very expensive, inducing a complexity of $O(n_e^3)$, where $n_e$ is the number of electrons. As a concrete example, DFT can be hundreds of thousands of times slower than a reasonably good GNN for inference [Gilmer et al., 2017].

*Active Learning (AL)* algorithms automatically identify the salient and exemplar samples from large amounts of unlabeled data [Settles, 2009, Ren et al., 2021]. This tremendously reduces the human annotation effort, as only the few samples identified by the algorithm need to be labeled manually. Further, since the deep network gets trained on the representative samples from the underlying data population, it typically depicts better generalization capability than a passive learner, where the training data are selected at random. Deep AL has been used with remarkable success in various applications, such as computer vision [Yoo and Kweon, 2019, Sinha et al., 2019], natural language processing [Zhang et al., 2022], medical diagnosis [Blanch et al., 2017], chemistry [Smith et al., 2018], and anomaly detection [Pimentel et al., 2020] among others. There are a few AL applications for 3D GNNs [Smith et al., 2021, van der Oord et al., 2023]; however, these works do not specifically account for 3D geometric information. The 3D geometry of molecules is crucial for determining molecular properties, but it introduces unique challenges in designing effective AL schemes. Currently, a principled AL algorithm for 3D molecular graphs is still lacking.

In this paper, we propose a principled AL paradigm for 3D molecular graphs. We formulate a criterion based on uncertainty and diversity, which ensures that the queried molecules are those where the graph learning model has maximal uncertainty about the labels, and that are also mutually diverse to avoid duplicate sample queries. In particular, diversity computing for 3D graphs is challenging and the *difficulties are twofold*. Firstly, the AL pipeline requires computing the difference between any two 3D molecular graphs, which could have different planar (2D) molecules (entangling different atom numbers, *etc.*), in most cases. Secondly, the 3D shape (geometry) of a 3D graph should be captured completely for expressive geometric representations and accurate diversity computation.

To tackle these challenges, we propose a novel diversity sampling method for 3D molecular graphs based on distributions of important 3D geometries. We propose a set of new 3D graph isometries for geometric modeling, which produces geometric representations that are at least as powerful as the Geometric Weisfeiler-Leman (GWL) test [Joshi et al., 2023] in distinguishing 3D graph geometries. This indicates our approach sets an upper bound on the expressive power of any existing 3D GNN models. Hence, the geometries derived from our geometric modeling method (*e.g.*, reference distances, triangles) can be used for accurate diversity computing. To compare any two 3D molecules (with different planar graphs), the moments of the distributions of the derived geometries are extracted for final diversity computing of 3D graphs. In addition, to ensure our AL paradigm selects samples with maximal uncertainties, we carefully design a Bayesian geometric GNN specifically for 3D graph uncertainty computing. Our method is shown to be effective and efficient based on a set of ground approximations. With our novel components, we pose the sample selection as a quadratic programming (QP) problem and implement a fast QP solver to identify exemplar molecules to be annotated. Our method is easy to implement and can be applied in conjunction with any 3D GNN architecture.

Overall, our proposed AL paradigm incorporates both diversity and uncertainty for 3D molecular graphs. The diversity component, driven by proposed geometric isometries, captures diverse chemical properties from geometries. The uncertainty component leverages chemical contexts, such as atom types, as node features, enhancing the model's ability to identify and learn from uncertain chemical interactions. By considering both, our method represents a powerful AL paradigm for 3D molecular graphs. We conduct extensive experiments, and the results demonstrate the effectiveness of the proposed diversity and uncertainty methods as well as the overall AL paradigm.

**Our contributions are summarized below.** (i) We propose a principled AL paradigm to alleviate the annotation hurdle of 3D molecular graphs. We employ diversity and uncertainty measures to select the most informative subset for AL. (ii) We introduce a novel diversity component for 3D molecular graphs. Investigating geometric graph isomorphism, we introduce a *model-agnostic* geometric modeling method, which is provably at least as expressive as the GWL test. Our method can significantly enhance the accuracy of diversity computing for 3D molecular graphs. (iii) Our proposed graph isometries set the theoretical upper bound to the expressive power of all existing 3D GNNs, and thus can serve as the new gold standard to test the expressiveness of various 3D

GNNs. (iv) Rooted in Bayesian inference, we develop an effective and efficient pipeline to compute uncertainties for 3D molecular graphs. (v) Our framework significantly outperforms mainstream AL baselines, achieving remarkable efficiency owing to the cheap complexity of $O(N^2)$ as well as the implementation of a fast QP solver.

## 2  Methods

### 2.1  Diversity Computing for 3D Molecular Graphs

In molecular AL tasks, diversity sampling is important for eliminating redundancy, thereby wisely leveraging the annotation budget. The model's capability of capturing the 3D shape diversity among molecules is crucial for informed sampling. A particular challenge is that a diversity measure for two 3D molecules with different planar graphs is indispensable. Methods for diversity measures for 3D molecules with the same planar graph have been developed [Kumar and Zhang, 2018, Kearnes et al., 2016, Gfeller et al., 2013], but a diversity method for two 3D molecules with different planar graphs (entailing different atoms, *etc*) is demanding. Inspired by the USR method [Ballester and Richards, 2007], we propose a novel solution to achieve the goal from the distribution perspective. Generally, we develop a set of new *isometries* for expressive representations of 3D molecular graphs, after which the distributions of geometries associated with the isometries are obtained for diversity computing.

#### 2.1.1  Isometries of 3D Molecular Graphs

As the first step, we introduce a set of new *isometries* as a basis, aiming at expressive representations of 3D graphs. As we focus on 3D geometry of molecules in this section, for simplicity, we use 3D point clouds to illustrate our ideas. Let $A = \{a_1, a_2, ..., a_n\}$ and $B = \{f(a_1), f(a_2), ..., f(a_n)\}$ be two sets representing 3D point clouds. Here, each $a_i$ in $A$ is associated with a positional vector $\boldsymbol{a_i} = (x_{a_i}, y_{a_i}, z_{a_i})$ in 3D space. $f$ denotes a bijective mapping between $A$ and $B$. Then, similarly, each point $f(a_i)$ in $B$ is associated with a positional vector $\boldsymbol{f(a_i)} = (x_{f(a_i)}, y_{f(a_i)}, z_{f(a_i)})$.

Two 3D point clouds, $A$ and $B$, are said to be $E(3)$-isomorphic, if there exists $\gamma \in E(3)$ such that $A = \gamma B$. We further choose or compute a consistent reference point (*e.g.*, centroid) for each point cloud, denoted as $r_1$ and $r_2$, respectively. Without loss of generality, we use $a_{\text{far}}$ to denote the farthest point from the reference point in point cloud $A$. Below, we will define three levels of isometries, each of which fulfills an isometric mapping between $A$ and $B$. To satisfy *any* isometry, there needs to exist a bijective function $f : A \to B$, such that $\boldsymbol{h_{f(a)}} = \boldsymbol{h_a}$ for any node $a \in A$. Here, $\boldsymbol{h_{f(a)}}$ and $\boldsymbol{h_a}$ denote the node feature vectors for $f(a)$ and $a$, respectively.

**Reference Distance Isometry:** If there exists a collection of global group elements $\gamma_i \in E(3)$, such that $(r_2, f(a_i)) = (\gamma_i r_1, \gamma_i a_i)$ for each point $a_i \in A$, $A$ is reference distance isometric to $B$.

Reference distance isometry involves the Euclidean distance between any atom in the molecule and the predefined reference point.

**Triangular Isometry:** If there exists a collection of global group elements $\gamma_i \in E(3)$, such that $(r_2, f(a_{\text{far}}), f(a_i)) = (\gamma_i r_1, \gamma_i a_{\text{far}}, \gamma_i a_i)$ for each point $a_i \in A$, $A$ is triangular isometric to $B$.

Figure 1: The illustrations of encoding the molecular triangular and cross-angular isometries

With reference point $r$, we define the reference vector $\boldsymbol{v_0}$ as $r$ pointing to the farthest point $a_{\text{far}}$ in a 3D molecule. Based on reference distance isometry, triangular isometry further involves the angle between $\boldsymbol{v_0}$ and other vectors pointing from $r$ to any other point in the molecule, computed as $\theta_k = \cos^{-1}\left(\frac{\boldsymbol{v_0} \cdot \boldsymbol{v_k}}{\|\boldsymbol{v_0}\| \|\boldsymbol{v_k}\|}\right)$, where $\boldsymbol{v_k}$ denotes vectors originating from $r$ and directed towards $k^{\text{th}}$ atoms in the molecule. The process is illustrated in part A of Fig. 1. For a molecule with $N$ nodes, we compute $N - 1$ angles. Essentially, such angles provide insights into the spatial arrangement of atoms with respect to the pre-assigned reference vector.

**Cross-angle Isometry:** If there exists a collection of global group elements $\gamma_{ij} \in E(3)$, such that $(r_2, f(a_j), f(a_i)) = (\gamma_{ij} r_1, \gamma_{ij} a_j, \gamma_{ij} a_i), \forall a_i, a_j \in A \ (i \neq j)$, $A$ is cross-angle isometric to $B$.

Beyond the angles in triangular isometry as well as based on reference distance isometry, cross-angular isometry further considers angles formed by any two atoms in the molecule with respect to

the reference vector as above. Specifically, for every pair of atoms $i$ and $j$, a vector $\boldsymbol{v}_{ij}$ is formed from $i$ to $j$. With the reference vector $\boldsymbol{v}_0$, the cross angle is computed as $\alpha_{ij} = \cos^{-1}\left(\frac{\boldsymbol{v}_0 \cdot \boldsymbol{v}_{ij}}{\|\boldsymbol{v}_0\|\|\boldsymbol{v}_{ij}\|}\right)$. This approach, as depicted in part B of Fig. 1, essentially reflects cross-angle information globally. For a molecule with $N$ nodes, we compute $N(N-1)/2$ cross angles with the complexity of $O(N^2)$.

Next, we propose **Theorem 1** to indicate the relationship between these three isometries as below.

**Theorem 1.** *If $A$ and $B$ are triangular isometric, then $A$ and $B$ are reference distance isometric; If $A$ and $B$ are cross-angle isometric, then $A$ and $B$ are triangular isometric.*

The proof of **Theorem 1** can be found in Appendix A.1. Generally, we define three levels of isometries for graph isomorphism. *Reference distance isometry*

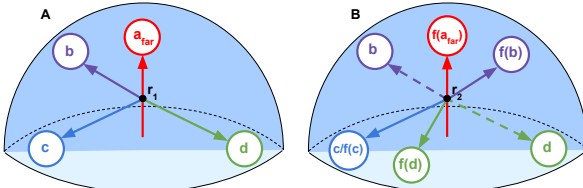

Figure 2: $A$ and $B$ are triangular isometric but not cross-angular isometric. The angles $\angle br_1 a_{far}$, $\angle cr_1 a_{far}$, and $\angle dr_1 a_{far}$ in structure $A$ are equal to the angles $\angle f(b)r_2 f(a_{far})$, $\angle f(c)r_2 f(a_{far})$, and $\angle f(d)r_2 f(a_{far})$ in structure $B$, respectively. However, the cross angle $\angle dr_1 c$ is not equal to the cross angle $\angle f(d)r_2 f(c)$.

ensures that the Euclidean distance between each point and a predefined reference point is consistent in two different point clouds. *Triangular isometry* further manifests the spatial arrangement of atoms referring to the pre-assigned pivot. Built on *triangular isometry*, *cross-angular isometry* then reflects the pair-wise global information. An illustrative example for *triangular isometry* and *cross-angular isometry* is also given in Fig. 2. Clearly, cross-angular isometry represents the strictest isometry among the three. In the following Sec. 2.1.2, we show that a designed geometric representation based on *cross-angular isometry* can exhibit great expressive power.

### 2.1.2 Expressive Power of Our Geometric Representations

In this section, we aim to formally elucidate the expressive power of a geometric representation (GR) based on our developed isometries in Sec. 2.1.1. Naturally, we formulate $GR_{\text{ours}}$ as a set containing all reference distances, triangles, and cross angles in a 3D graph.

We explore the Geometric Weisfeiler-Leman (GWL) test [Joshi et al., 2023], and then leverage GWL to illustrate the expressiveness power of our model. GWL test is an extension of the classic WL Test, enhancing its capabilities by incorporating both the topological structure of the graph and the geometric attributes of its vertices. Such an integration allows the GWL test especially apt for evaluating all 3D graph representation methods. Similar to the regular WL test, GWL test imposes an upper bound to the expressive power of 3D GNNs, *i.e.*, if GWL test fails to distinguish two 3D graphs, then all existing 3D GNNs would also fail. See details of the GWL test in Appendix A.2.

**Proposition 1.** *$GR_{ours}$ is at least as expressive as the GWL test. In other words, $GR_{ours}$ suffices to distinguish any non-isomorphic molecular structures that are distinguishable by any 3D GNN.*

The proof of **Proposition 1** can be found in Appendix A.1. In conclusion, the molecular geometric representation $GR_{\text{ours}}$ developed in this work has the greater expressive power than the GWL test, which indicates our diversity sampling method is accurate enough to capture the 3D shape diversity among different molecules. Notably, as mentioned before, GWL test sets the upper bound to the expresiveness of any existing 3D GNNs. ***Apparently, our geometric representation $GR_{ours}$ is provably at least as powerful as any existing 3D GNN for learning geometric features.*** Essentially, the three isometries associated with $GR_{\text{ours}}$ define expressiveness at different levels. For example, as only considering distance information, a well pretrained SchNet is upper bounded by reference distance isometry (but not triangular isometry or cross-angular isometry); as a more powerful model than SchNet, a well pretrained DimeNet is upper bounded by triangular isometry (but not cross-angular isometry). Additionally, learning accurate geometric representations requires a perfectly pretrained 3D GNN model, which is hard to guarantee in practice. ***Our isomorphy study provides a deterministic and model-agnostic diversity component for 3D graphs, avoiding the need of a 'perfectly' pretrained 3D GNN model, as well as achieving a theoretically guaranteed upper bound of the expressiveness of all existing 3D GNN models.***

### 2.1.3 Final Distributional Representations

Based on the isomorphy study in Sec. 2.1.1, we obtain our geometric representation method $GR_{\text{ours}}$ and prove $GR_{\text{ours}}$ possesses greater expressive power than any existing 3D GNN models in Sec.

2.1.2. In this section, we aim to extract the distributions of the ***entangled three geometries in*** $GR_{ours}$***, including reference distances, triangles, and cross angles***, for diversity computing. Fortunately, we have the theorem [Hall, 1983] implying that the sequence of translated moments can be used to determine the original distribution. Following the USR work [Ballester and Richards, 2007], for each of the three aforementioned geometries, we also use four reference points to reflect the "translated" geometries; those are, the centroid (denoted as ctd) computed by the mean position of all the atoms in the 3D molecule, the point closest to the centroid (denoted as cst), the point farthest from the centroid (denoted as fct), and the point farthest from fct (denoted as ftf). For each reference point, we use a set of moments, including mean, variance, skewness, and kurtosis, which describe a distribution from different angles, *e.g.*, skewness indicates the asymmetry and kurtosis describes the tailedness of a distribution. Detailed formulae for these moments can be found in the Appendix A.3. Notably, we compute these translated moments for all three entangled geometries as above. Eventually, we obtain summarized representations of distributions over geometries of 3D graphs, capturing essential characteristics of a molecule's shape.

We use cross angles as an example to describe the final distributional vector. For a molecule with $N$ atoms, as shown in Fig. 1, we can obtain a set of cross angles $[\alpha_{ij}^{\text{ref}}]_{i \neq j, 0 < i, j < N}^{N(N-1)/2}$ for a reference point (*e.g.*, ctd). After applying statistical moments as an approximation, we can obtain a 4-dimensional vector $\overrightarrow{M_{\text{ref}}^{\text{ca}}} = [m_{\text{ref}}^{\text{ca}}, v_{\text{ref}}^{\text{ca}}, s_{\text{ref}}^{\text{ca}}, k_{\text{ref}}^{\text{ca}}]$, where the four elements denote the mean, variance, skewness, and kurtosis for this reference point, respectively. We perform a similar process for all four reference points mentioned above. By doing this, we can obtain four 4-dimensional vectors including $\overrightarrow{M_{\text{ctd}}^{\text{ca}}}$, $\overrightarrow{M_{\text{cst}}^{\text{ca}}}$, $\overrightarrow{M_{\text{fct}}^{\text{ca}}}$, and $\overrightarrow{M_{\text{ftf}}^{\text{ca}}}$, which are then concatenated together, resulting in the final 16-dimensional vector to represent the distribution of cross angles. We repeat the similar process for reference distances and triangles, and then all three corresponding 16-dimensional vectors are further concatenated as a 48-dimensional distributional vector to represent the geometric information of the input molecule. The 48-dimensional distributional vectors are then used to compute the diversity matrix. For any two molecules $n_1$ and $n_2$ in the dataset with $N$ molecules, we perform the inner product on their distributional vectors to achieve the similarity, and then use $1-$ similarity to obtain the final value $D_{n_1 n_2}$ as the diversity measure between them. Finally, a matrix $D \in \Re^{N \times N}$ is obtained, which contains the diversity between every pair of molecules.

**Comparing Our Method to Traditional Structural Descriptors.** Our method generates a 48-dimensional vector that encodes the geometric structure of a molecule. This representation is both equivariant to roto-translations and invariant to atomic permutations as the statistical quantities remain unchanged under such transformations. In contrast, Smooth Overlap of Atomic Positions (SOAP) [Bartók et al., 2013, De et al., 2016, Jäger et al., 2018] generates atom-wise vectors that capture local atomic environments by employing spherical harmonics and radial basis functions. While SOAP is also equivariant to roto-translations, it is not invariant to atomic permutations. On the other hand, Atomic Cluster Expansion (ACE) [Drautz, 2019] uses a systematic expansion to describe interactions of varying orders (*e.g.*, two-body, three-body interactions). However, ACE is less of a traditional descriptor compared to our method and SOAP; it is designed to provide a complete and systematic representation of atomic interactions by focusing on higher-order expansions (e.g., two-body, three-body interactions). This makes ACE more comprehensive in capturing the physical interactions within a system, but less suited for producing a fixed-dimensional, flexible descriptor. Unlike our method and SOAP, which generate more compact and adaptable descriptors, ACE emphasizes thorough expansions, making it less ideal for tasks requiring flexible, low-dimensional representations that can adapt easily to the active learning scheme. An empirical comparison between our method and the approach that uses SOAP will be provided, highlighting the effectiveness of our method in capturing molecular geometries.

## 2.2 Uncertainty Computing for 3D Molecular Graphs

In Sec. 2.1, we develop an effective method for diversity computing among different 3D molecular graphs. In addition to selecting diverse molecules, it is important to select molecules where the model has maximal prediction uncertainty about the labels, so as to append maximal information to the model. Uncertainty qualification is well-studied in planar graph analysis [Hirschfeld et al., 2020], but an effective paradigm for 3D molecular graphs is currently lacking. Additionally, existing methods, such as Bayesian neural networks (BNNs) [Lampinen and Vehtari, 2001, Titterington, 2004, Goan and Fookes, 2020] and deep model ensemble methods [Lakshminarayanan et al., 2017, Huang et al., 2017], are excessively computationally expensive, limiting their capacity in 3D graph analyses. In a

concurrent work [Thaler et al., 2024] on active learning for partial charge prediction of metal-organic frameworks, a dropout Monte Carlo scheme has been proposed to lessen these issues.

In this work, we develop an effective and efficient method, known as Bayesian geometric graph neural network (BGGNN), that takes a 3D graph as input and produces the demanding properties as well as uncertainty values, *e.g.*, mean and variance. Formally, a 3D graph is represented as $\mathbf{G} = (V, E, P)$, where $V$ denotes the set of vertices (atoms), $E$ denotes the set of edges (bonds), and $P$ denotes the set of Cartesian coordinates for all atoms. A 3D molecular graph is associated with a set of properties, denoted as $\mathbf{O}$. Recently, researchers have developed 3D GNNs, such as SchNet [Schütt et al., 2017], DimeNet [Gasteiger et al., 2020b], SphereNet [Liu et al., 2022], and GemNet [Gasteiger et al., 2021], for 3D graph representation learning. The likelihood of a 3D GNN can be represented as $p_{\text{3DGNN}}(\mathbf{O} \mid \mathbf{G}, \mathbf{w})$, where 3DGNN indicates any existing 3D GNN and $\mathbf{w}$ denotes the set of parameters of the used 3D GNN. We also use $p_{\text{3DGNN}}(\mathbf{w})$ to represent the prior distribution for the parameters. Assume we collect a new input and output pair, denoted as $\mathbf{g}^*$ and $\mathbf{o}^*$. Then based on the conventional Bayesian theorem, Bayesian inference for this new output $\mathbf{o}^*$ is given by

$$p_{\text{3DGNN}}\left(\mathbf{o}^* \mid \mathbf{g}^*, \mathbf{G}, \mathbf{O}\right) = \int_{\mathbb{R}^n} p_{\text{3DGNN}}\left(\mathbf{o}^* \mid \mathbf{g}^*, \mathbf{w}\right) p_{\text{3DGNN}}(\mathbf{w} \mid \mathbf{G}, \mathbf{O}) d\mathbf{w}, \tag{1}$$

where $\mathbb{R}^n$ is the whole space of $n$ parameters in 3DGNN. It's infeasible to perform the above integration on $\mathbb{R}^n$ due to prohibitive computational cost. To tackle this, the variational inference method is introduced to approximate $p_{\text{3DGNN}}(\mathbf{O} \mid \mathbf{G}, \mathbf{w})$ with the parameterized $q_\theta(\mathbf{w})$ through minimizing the Kullback-Leibler (KL) divergence between these two distributions. After applying Bayesian theorem once more, the minimization objective becomes

$$\mathcal{L}_{\text{VI}}(\theta) = -\int_{\mathbb{R}^n} q_\theta(\mathbf{w}) \log p_{\text{3DGNN}}(\mathbf{O} \mid \mathbf{G}, \mathbf{w}) d\mathbf{w} + \text{KL}\left(q_\theta(\mathbf{w}) \| p_{\text{3DGNN}}(\mathbf{w})\right), \tag{2}$$

To completely avoid the integration over the whole parameter space, the MC-dropout method [Gal and Ghahramani, 2016, Srivastava et al., 2014] is further used in our BGGNN. Specifically, it employes the Monte-Carlo estimator [Gal et al., 2016, Gal and Ghahramani, 2016] to approximate the integration by performing summation over the sampled models. In practice, researchers implement an MC-dropout network by using dropout as the network's regularization[Gal and Ghahramani, 2016]. Following this, we propose to insert dropout layers after the linear layers in our used 3DGNN as an effective yet efficient estimation of Bayesian inference.

Now as we have obtained the variational predictive distribution of a new output with $q_\theta(\mathbf{w})$, we can easily compute the predictive mean and variance of this distribution. For the molecular property prediction tasks, after we sample $N$ outputs from the same input, the heteroscedastic predictive uncertainty is then given by

$$\widehat{\sigma^2}\left(\mathbf{o}^* \mid \mathbf{g}^*\right) = \frac{1}{N} \sum_{n=1}^{N} (\hat{\mathbf{o}}_n^*)^2 - \left(\frac{1}{N} \sum_{n=1}^{N} \hat{\mathbf{o}}_n^*\right)^2 + \frac{1}{N} \sum_{n=1}^{N} \widehat{\sigma}_n^2, \tag{3}$$

where $\hat{\mathbf{o}}_n^*$ is the $n^{th}$ sampled output and $\widehat{\sigma}_n^2$ is the variance that is the same among all the data samples. By doing this, we can obtain an uncertainty value (variance) for each molecule. Additionally, built on a 3D GNN, our BGGNN can faithfully produce a set of molecular properties $\mathbf{O}$.

Practically, any of the existing 3D GNN can be used as the backbone network for property prediction and uncertainty computing. In this study, we employ SphereNet [Liu et al., 2022] as our 3DGNN, owing to its great power in incorporating 3D geometric information. We apply dropout layers onto the linear layers of SphereNet for Bayesian inference in our BGGNN. To allow more accurate AL selections, we particularly employ the concrete dropout with a learnable dropout rate [Gal et al., 2017] in our BGGNN. Overall, our method is shown to be an effective and efficient paradigm for 3D graph uncertainty computing, as further empirically demonstrated in Sec. 4.

## 2.3 Active Sampling

A schematic diagram of our active sampling framework is depicted in Fig. 6 and described in A.4 in Appendix. Specifically, in Sec. 2.1, we obtain the matrix $D \in \Re^{N \times N}$ containing the mutual diversity between every pair of unlabeled molecules, where $N$ is the number of unlabeled molecules. In Sec. 2.2, we employ our designed BGGNN to achieve the vector $r \in \Re^{N \times 1}$ quantifying the prediction uncertainty score of each unlabeled molecule. In the AL setting, our objective is to select a batch of $k$ unlabeled molecules ($k$ is a

$$\begin{aligned} \max_z \quad & z^\top r + \lambda z^\top D z \\ s.t. \quad & \sum_{i=1}^N z_i = k \\ & z_i \in \{0, 1\}, \forall i, \quad (4) \end{aligned}$$

pre-defined query batch size) with high prediction uncertainty and high mutual diversity among them. Let $z \in \{0, 1\}^{N \times 1}$ be a binary vector with $N$ entries which denotes whether the unlabeled molecule $x_i$ will be included in the batch ($z_i = 1$) or not ($z_i = 0$). The molecule selection can thus be posed as the following optimization problem as in Eq. (4), where $\lambda$ is a weight parameter governing the relative importance of the two terms. This is a standard quadratic programming (QP) problem; we relax the integer constraints into continuous constraints and solve the problem using an off-the-shelf QP solver. In this work, we employ the widely used Operator Splitting Quadratic Program (OSQP) [Stellato et al., 2020] to solve the QP problem in Eq. (4). We then apply a greedy approach to project the continuous solution back to the binary space, where the $k$ highest entries of the continuous solution vector are set to 1 and the remaining to 0. Such an approach is commonly used to convert continuous solutions obtained from a QP solver to binary solutions in AL [Chattopadhyay et al., 2013, Wang and Ye, 2013]. To accelerate the optimization, we implement a solution to execute the problem in the GPU (instead of the CPU) using the parallel implementation of the alternating direction method of multipliers, as detailed in Schubiger et al. [2020]. Notably, the predictions in the main tasks (*e.g.*, molecular properties) are produced by our BGGNN built on SphereNet as in Sec. 2.2.

## 3 Related Work

### 3.1 Active Learning

AL is a well-researched problem in the machine learning community [Settles, 2009]. There exist two commonly used strategies for AL sampling. Uncertainty based sampling queries unlabeled samples with the highest prediction uncertainties for annotation. Diversity/representativeness based sampling aims to select the subset that can well represent the entire data distribution. A full review of the two AL sampling methods is provided in Appendix A.5.

### 3.2 Molecular Shape Similarity

Molecular shape similarity plays a pivotal role in drug discovery and virtual screening of compounds [Kumar and Zhang, 2018, Murgueitio et al., 2012, Shang et al., 2017]. Methods predominantly fall into several categories [Kumar and Zhang, 2018], including descriptor-based methods [Schreyer and Blundell, 2012, Cannon et al., 2008, Li et al., 2016, Armstrong et al., 2009, Zhou et al., 2010], atom-centered Gaussian-based methods [Haque and Pande, 2010, de Lima and Nascimento, 2013, Yan et al., 2013], surface-based methods [Hofbauer et al., 2004, Mavridis et al., 2007, Cai et al., 2012, Karaboga et al., 2013, Venkatraman et al., 2009, Sael et al., 2008], *etc*. Descriptor-based methods are notably represented by the Ultrafast Shape Recognition (USR) algorithm [Ballester and Richards, 2007], which uses statistical moments of the distance distribution to characterize molecular shapes. Gaussian overlay-based methods, with ROCS [Rush et al., 2005, Hawkins et al., 2007] being the most commonly used one, evaluate the maximum volume overlap between two molecules. Surface-based methods typically employ shape signatures [Zauhar et al., 2013] or shape histograms to delineate molecular surfaces for shape similarity assessment. Despite the progress, a principled and theoretically ground similarity method for 3D molecular graphs is currently lacking.

## 4 Experiments

### 4.1 Experimental Setup

**Implementation Details**: We use two mainstream 3D GNNs SphereNet [Liu et al., 2022] and DimeNet$^{++}$ [Gasteiger et al., 2020a] as the backbone models of our BGGNN. We directly use the optimal network configurations from the original papers for both backbone models. We train the network for 200 epochs, unless otherwise specified. We use the *Adam Optimizer* with an initial learning rate $5 \times 10^{-4}$ and scale it by a factor of 0.5 every 15 epochs.

**Data and Active Learning Setup**: We first perform experiments on the QM9 benchmark dataset. Since SphereNet is more stable and incorporates more 3D information, we conduct experiments on *mu, alpha, homo, and lumo* for SphereNet, and *mu and lumo* for DimeNet$^{++}$. These properties have continuous values, making the prediction problem a regression task. We randomly divide the training set of $110,000$ molecules into three splits of size $25,000$ each. From each split, we randomly select $5,000$ molecules as the initial labeled set and the remaining $20,000$ molecules as the unlabeled set. In each AL iteration, we query $1,500$ molecules from the unlabeled set, which are labeled and appended to the labeled set. The model's performance is evaluated on a held-out validation set containing $10,000$ molecules. We save the best-performing model on the validation set and report its performance on the test set containing $10,831$ molecules. The process is repeated for 7 AL iterations, which is taken as the stopping criterion. The final results are averaged over the three splits to rule out

the effects of randomness. $\lambda$ in Eq. 4 is taken as 1. The Mean Absolute Error (MAE) is used as the evaluation metric. In addition, to study the generalizability of our framework to more geometric data, we also conduct experiments to predict atomic forces for **Aspirin** in MD17 using our framework.

**Comparison Baselines**: We use four classic AL methods as baselines: *Random Sampling*, *Coreset* [Sener and Savarese, 2018], *Learning Loss* [Yoo and Kweon, 2019], and *Evidential Uncertainty* [Beluch et al., 2018, Amini et al., 2020]. *Random Sampling* is the default comparison baseline in AL research. *Coreset* and *Learning Loss* are two extensively used deep active learning algorithms for regression applications. *Evidential Uncertainty* is also a commonly used technique to quantify uncertainty for molecular property prediction and was hence included as a comparison baseline. Note some existing studies [Kulichenko et al., 2023, Gusev et al., 2023, Craig and García-Melchor, 2021] have applied AL to molecule research and chemistry. However, these works focus on 2D molecules without considering 3D geometry, which is the focus of our work. Additionally, the techniques used in existing studies can arguably fall into the aforementioned AL categories. Hence, we think comparing with these classic AL methods is sufficient to demonstrate the superiority of our pipeline.

## 4.2 Active Learning Performance

The active learning performance with SphereNet is depicted in Fig. 3. In each graph, the $x$-axis denotes the iteration number and the $y$-axis denotes the MAE on the test set. Our analysis revealed that *Evidential Uncertainty* depicted the worst performance and furnished significantly high error values for all four properties, which obscured the difference in performance among the other methods in the plots. For better interpretation and understanding, we exclude the *Evidential Uncertainty* method from the plots here and present the results with this baseline in Sec. A.6 of the Appendix. The other baseline methods depict more or less similar performance, with *Coreset* marginally outperforming the other baselines. Our method comprehensively outperforms all the baselines. At any given AL iteration, it consistently attains a lower MAE compared to all the baselines.

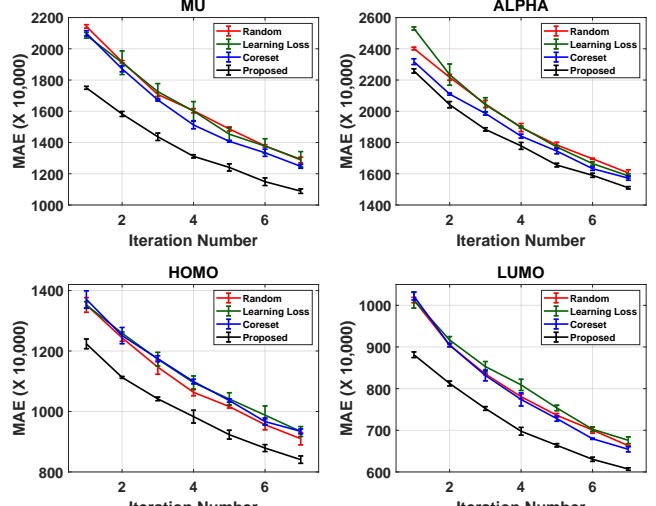

Figure 3: Active learning performance results with SphereNet. The graphs show the mean (averaged over 3 runs) and the error-bars for all the methods. We plot the MAE values from the first iteration onwards, to focus on the comparative performance of the methods after they start selecting samples using AL. Best viewed in color.

We also conducted statistical tests of significance using paired t-test to assess whether the improvement in performance achieved by our method is statistically significant. For this purpose, we compared the average MAE achieved by our method against each of the baselines individually. The results are reported in Table 1; each entry in the table denotes the p-value of the paired t-test between our method

Table 1: The table shows the p-values obtained using paired t-test between the results our method against each of the baselines for all the properties studied. *Here, L. Loss refers to Learning Loss.*

| Properties | Baselines | | | |
|---|---|---|---|---|
| | Random | L. Loss | Coreset | Evidential |
| *mu* | $7.54 \times 10^{-6}$ | $5.09 \times 10^{-5}$ | $1.51 \times 10^{-4}$ | $2.19 \times 10^{-7}$ |
| *alpha* | $1.06 \times 10^{-5}$ | $8.14 \times 10^{-4}$ | $4.27 \times 10^{-5}$ | $2.72 \times 10^{-4}$ |
| *homo* | $2.26 \times 10^{-5}$ | $8.36 \times 10^{-7}$ | $4.23 \times 10^{-6}$ | $1.71 \times 10^{-8}$ |
| *lumo* | $4.48 \times 10^{-5}$ | $1.25 \times 10^{-5}$ | $3.12 \times 10^{-4}$ | $2.39 \times 10^{-6}$ |

and the corresponding baseline (denoted in the columns) for the property studied (denoted in the rows). From the table, we note that the improvement in performance achieved by our method is statistically significant ($p < 0.05$) compared to all the baselines, consistently for all the four properties studied. These results unanimously corroborate the promise and potential of the proposed active sampling method to tremendously reduce the annotation cost in inducing a robust 3D graph neural network for molecular property prediction.

In addition, to study the robustness of our framework to the underlying network architecture and generalizability to the underlying geometric graph data, we have the following results: *1. To study the*

***robustness of our framework to the underlying network architecture***, results on the ***mu and lumo*** properties of the QM9 dataset using DimeNet$^{++}$ [Gasteiger et al., 2020a] as the backbone model are presented in Section A.7 of the Appendix due to space constraints. The results depict a similar pattern as Figure 3, with the proposed method consistently outperforming all the baselines for both the properties. A paired t-test, presented in Table 3 revealed that the performance improvement achieved by our framework is statistically significant. ***2. To study the generalizability of our framework to the underlying geometric graph data***, results on predicting atomic forces for ***Aspirin*** molecules in the MD17 benchmark dataset [Chmiela et al., 2017b] using our framework are depicted in Section A.8 of the Appendix and further corroborate the potential of our framework.

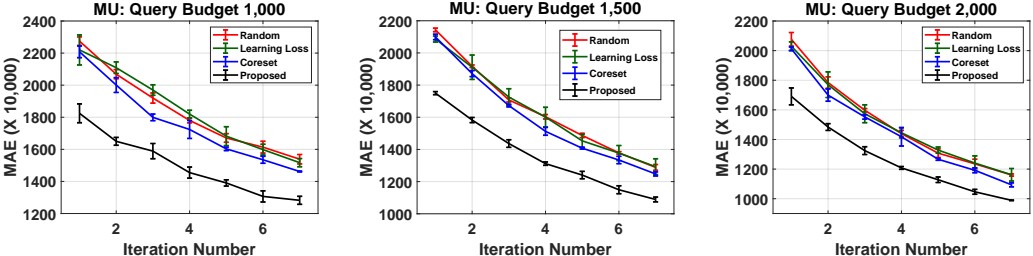

Figure 4: Study of query budget on the active learning performance. The graphs show the mean (averaged over 3 runs) and the errorbars for all the methods. The results with budget 1500 are the same as the those presented in Figure 3 and are included here for comparison. Best viewed in color.

## 4.3 Study of Query Budget

The goal of this experiment is to study the effect of query budget (batch size) on the AL performance. The results on the ***mu*** property with SphereNet for query budgets $1,000$, $1,500$ and $2,000$ are depicted in Fig. 4. Since *Evidential Uncertainty* depicted much worse perfor-mance than all the methods, it

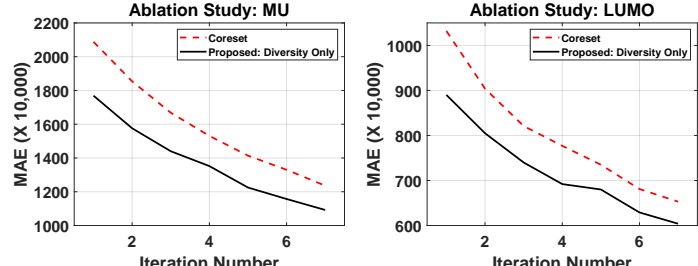

Figure 5: Ablation study results on the ***mu*** and ***lumo*** properties with SphereNet. Best viewed in color.

was excluded from this comparison. Our framework once again outperforms all the baselines consistently for all the query budgets. As before, we conducted a paired t-test and the results are presented in Appendix A.9. From the p-values, we conclude that the error values furnished by our method are statistically significantly better ($p < 0.05$) than all the baselines, consistently for all the query budgets. These results are particularly significant from a practical standpoint as the available query budget in a real-world application is dependent on time, resources, and other constraints.

## 4.4 Ablation Studies

We conduct ablation studies to examine the power of our diversity computing method, as it is our primary contribution in this research. We perform experiments on the ***mu*** and ***lumo*** properties with SphereNet from two aspects. Firstly, we compare our framework with only the diversity term in Eq. 4 against *Coreset*, the state-of-the-art diversity-based AL technique. The results are reported in Fig. 5, from which we note that the diversity component of our framework consistently furnishes much lower MAE values than *Coreset* over all the AL iterations, for both properties. Secondly, we also conducted experiments where we compared the performance of our overall framework (using both uncertainty and diversity) against the baseline where only the uncertainty term in Eq. (4) was used for active sampling. The results revealed that removing the diversity term adversely affected the performance of our framework. A paired t-test revealed that the improvement in performance achieved by our diversity component is statistically significant ($p < 0.05$) for both these properties ($p = 0.0001$ for ***mu*** and $p = 0.04$ for ***lumo***). These results show the effectiveness of the proposed diversity metric for AL framework to train a 3D GNN for molecular property prediction. Additionally, we examine the individual impact of diversity and uncertainty components in Appendix A.10. We also compare our proposed diversity component with the SOAP-based diversity, and test our method against BatchBALD [Kirsch et al., 2019], a greedy clustering-based Bayesian uncertainty approach in Appendix A.10.

## 4.5 Computation Time Analysis

In this experiment, we analyze the computation time of all the methods studied in this paper. The average time taken to query a batch of unlabeled samples and update the SphereNet model (one active learning iteration) are shown in Table 2. For fair comparison, all the methods were run on the same NVIDIA RTX A4500 20GB GPU.

The computation time of our framework is much less than *Coreset*, which needs to solve a mixed integer programming (MIP) problem. The other three methods have similar

Table 2: Average ($\pm$ std) time (minutes) taken by each method for sample selection and training the SphereNet model (one iteration of AL). *Here, L. Loss refers to Learning Loss.*

| Random | L. Loss | Coreset | Evidential | Ours |
|---|---|---|---|---|
| $53 \pm 4.5$ | $56 \pm 2.1$ | $127 \pm 3.5$ | $56 \pm 2.3$ | $64.9 \pm 7.5$ |

computation time, as they don't involve iterative algorithms. Our method takes **only slightly more** time than them, owing to the implementation of a faster QP solver as mentioned in Sec. 2.3, as well as our vectorized implementation to enable the use of GPUs to perform diversity matrix computation. The performance studies in Sec. 4.2 show that our framework is much more accurate than these baselines, and the ablation studies in Sec. 4.4 indicate both the diversity and uncertainty components are necessary to form a QP problem. Given the large margin of performance improvement, we think the efficiency of our method is acceptable.

## 5 Conclusion, Limitations, Future Work, and Broader Impacts

We present a principled active learning framework with the goal of reducing the annotation cost for learning from 3D molecules represented as 3D graphs. The sample selection is posed as a QP problem, which selects samples with high mutual diversity and high uncertainty. Novel diversity and uncertainty components are proposed for 3D graphs, with strong empirical results presented.

We present a model-agnostic diversity component for 3D graphs, and our method is provably at least as powerful as any existing 3D GNN for learning geometric information. Even though our method can set the upper bound of the accuracy of diversity sampling for 3D molecules, it remains unexplored if such an advantage can be incorporated into 3D GNN models for diversity sampling. For example, molecular similarity might be incorporated into 3D GNNs to achieve comparable AL performance. Moreover, our experimental studies focus on small molecules in this work.

As part of future work, we plan to apply our methods to problems where much more accurate but expensive annotation is required, such as computing molecular systems' ground states using the Schrödinger equation. DFT calculations are widely used but still involve approximations, as Schrödinger equation is prohibitively expensive and its use is limited in very small molecules. Our AL pipeline is anticipated to unleash greater potential in such extreme-scale applications. Additionally, given AL needs several interactions with each requiring the model is well-trained, we test our methods on the commonly used but medium-scale QM9 and MD17 datasets in this work. Even though we think the empirical studies are sufficient to support our theory, we still plan to test the scalability of our methods on large-scale molecule datasets, such as OC20 [Chanussot et al., 2021], in the future.

This work facilitates a new avenue in graph analysis by effective and efficient representation of 3D geometric information, thereby dramatically advancing graph learning and mining. Our methods can reduce the annotation cost for molecular data and also have the potential in a broad set of scientific data types, such as materials and proteins, facilitating various disciplines including basic biology, material science, and quantum chemistry. This work is anticipated to have strong impacts on drug discovery and material design by enabling low-cost representation learning. Any positive and negative societal impact associated with those applications and domains can be applied to our methods.

## 6 Acknowledgment

This work of Y. Liu has used the computational equipment supported by the US Army Research Office under the award W911NF-20-10159. Y. Liu and W. Gao would like to thank Xiaolin Li at Stony Brook University for providing this computational equipment. The work of S. Chakraborty is partially supported by the National Science Foundation under the award IIS-2143424 (NSF CAREER Award).

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

# A Appendix

## A.1 Proofs of the Theorems and Propositions

**Theorem 1.** *If $A$ and $B$ are triangular isometric, then $A$ and $B$ are reference distance isometric; If $A$ and $B$ are cross-angle isometric, then $A$ and $B$ are triangular isometric.*

*Proof.* Suppose that $A$ and $B$ are triangular isometric, then there exists a collection of global group elements $\gamma_i \in SE(3)$ such that

$$(r_2, f(a_{\mathrm{far}}), f(a_i)) = (\gamma_i r_1, \gamma_i a_{\mathrm{far}}, \gamma_i a_i), \quad \forall a_i \in A.$$

It follows immediately that for each point $a_i \in A$, $(r_2, f(a_i)) = (\gamma_i r_1, \gamma_i a_i)$ also holds. Thus, if $A$ and $B$ are triangular isometric, then $A$ and $B$ are reference distance isometric.

Suppose that $A$ and $B$ are cross-angular isometric, then there exists a collection of global group elements $\gamma_{ij} \in SE(3)$ such that

$$(r_2, f(a_j), f(a_i)) = (\gamma_{ij} r_1, \gamma_{ij} a_j, \gamma_{ij} a_i), \quad \forall a_i, a_j \in A, i \neq j.$$

By fixing $a_j$ to be $a_{far}$ and the corresponding $\gamma_{ij}$ to be $\gamma_i$, it follows immediately that for each point $a_i \in A$, there exists a collection of global group elements $\gamma_i \in SE(3)$ such that $(r_2, f(a_{\mathrm{far}}), f(a_i)) = (\gamma_i r_1, \gamma_i a_{\mathrm{far}}, \gamma_i a_i)$. Thus, if $A$ and $B$ are cross-angular isometric, then $A$ and $B$ are triangular isometric. $\qquad\square$

**Proposition 1.** *$GR_{ours}$ is at least as expressive as the GWL test. In other words, $GR_{ours}$ suffices to distinguish any non-isomorphic molecular structures that are distinguishable by any 3D GNN.*

*Proof.* We prove the case when the reference points $r_1$ and $r_2$ are the centroids of the point clouds $A$ and $B$, respectively. The proof for other choices of reference points follows analogously. First, we will show that $\zeta$, which is the function that gives us the geometric representation $GR_{\mathrm{ours}}$ given a point cloud, is $E(3)$-orbit injective.

Without loss of generality, assume that the centroids of these two point clouds are at the origin. Otherwise, they can be fixed by a translation in $\mathbb{T}(3) \cong E(3)/O(3)$. For simplicity, we denote the point $f(a_i)$ as $b_i$, and bold symbol represents the corresponding vectors. Suppose that $GR_{\mathrm{ours}}$ is the same for both points clouds $A$ and $B$, that is to say, we have the following conditions:

$$\|\boldsymbol{a_i}\| = \|\boldsymbol{b_i}\|, \ \forall i \in \mathbb{N}_{\leq n} \tag{5}$$

$$\frac{\langle \boldsymbol{a}_{\mathrm{far}}, \boldsymbol{a_i} \rangle}{\|\boldsymbol{a}_{\mathrm{far}}\| \cdot \|\boldsymbol{a_i}\|} = \frac{\langle \boldsymbol{b}_{\mathrm{far}}, \boldsymbol{b_i} \rangle}{\|\boldsymbol{b}_{\mathrm{far}}\| \cdot \|\boldsymbol{b_i}\|}, \ \forall i \in \mathbb{N}_{\leq n} \tag{6}$$

$$\frac{\langle \boldsymbol{a}_{\mathrm{far}}, \boldsymbol{a_i} - \boldsymbol{a_j} \rangle}{\|\boldsymbol{a}_{\mathrm{far}}\| \cdot \|\boldsymbol{a_i} - \boldsymbol{a_j}\|} = \frac{\langle \boldsymbol{b}_{\mathrm{far}}, \boldsymbol{b_i} - \boldsymbol{b_j} \rangle}{\|\boldsymbol{b}_{\mathrm{far}}\| \cdot \|\boldsymbol{b_i} - \boldsymbol{b_j}\|}, \ \forall i,j \in \mathbb{N}_{\leq n}, i \neq j. \tag{7}$$

It follows from (5) and (6) that for any $k \in \mathbb{N}_{\leq n}$,

$$\langle \boldsymbol{a}_{\mathrm{far}}, \boldsymbol{a_k} \rangle = \langle \boldsymbol{b}_{\mathrm{far}}, \boldsymbol{b_k} \rangle.$$

Then, for all $i,j \in \mathbb{N}_{\leq n}, i \neq j$,

$$\langle \boldsymbol{a}_{\mathrm{far}}, \boldsymbol{a_i} - \boldsymbol{a_j} \rangle = \langle \boldsymbol{b}_{\mathrm{far}}, \boldsymbol{b_i} - \boldsymbol{b_j} \rangle.$$

Thus, it is clear from (7) that all the pair-wise distances are the same, i.e., $\|\boldsymbol{a_i} - \boldsymbol{a_j}\| = \|\boldsymbol{b_i} - \boldsymbol{b_j}\|$ for all $i,j \in \mathbb{N}_{\leq n}, i \neq j$. Thus

$$\|\boldsymbol{a_i} - \boldsymbol{a_j}\|^2 = \|\boldsymbol{a_i}\|^2 - 2\langle \boldsymbol{a_i}, \boldsymbol{a_j} \rangle + \|\boldsymbol{a_j}\|^2$$
$$\|\boldsymbol{b_i} - \boldsymbol{b_j}\|^2 = \|\boldsymbol{b_i}\|^2 - 2\langle \boldsymbol{b_i}, \boldsymbol{b_j} \rangle + \|\boldsymbol{b_j}\|^2$$

It follows that $\langle \boldsymbol{a_i}, \boldsymbol{a_j} \rangle = \langle \boldsymbol{b_i}, \boldsymbol{b_j} \rangle$ from (5).

It is safe to assume that $(\boldsymbol{a_1}, \boldsymbol{a_2}, \ldots, \boldsymbol{a_n})$ spans $\mathbb{E}^3$, otherwise the proof is trivial when all points are co-planer. Without loss of generality, let $(\boldsymbol{a_1}, \boldsymbol{a_2}, \boldsymbol{a_3})$ be a basis for $\mathbb{E}^3$. It is easy to see that $(\boldsymbol{b_1}, \boldsymbol{b_2}, \boldsymbol{b_3})$ is also a basis for $\mathbb{E}^3$.

Let $X$ and $Y$ denote the matrices whose columns are $(a_1, a_2, a_3)$ and $(b_1, b_2, b_3)$, respectively. Let $G$ denote the associated Gram matrix, i.e. $G = X^T X = Y^T Y$, then $G$ is symmetric and positive semi-definite. Moreover, as both $X$ and $Y$ are full-rank, there exist orthogonal matrices $Q_X, Q_Y$ and upper triangular matrices $R_X, R_Y$ such that

$$\begin{cases} X = Q_X R_X \\ Y = Q_Y R_Y \end{cases}$$

then

$$\begin{cases} G = X^\top X = R_X^\top R_X \\ G = Y^\top Y = R_Y^\top R_Y \end{cases}$$

The form above follows the pattern of Cholesky decomposition. As $G$ is symmetric and positive semi-definite, the Cholesky decomposition is unique. Thus, $R_X = R_Y$. Thus, $X = Q_X Q_Y^{-1} Y$, where $Q_X Q_Y^{-1}$ is an orthogonal matrix. Thus, there exists $g \in O(3)$ such that $gX = Y$. If $n \leq 3$, this completes the proof.

When $n \geq 4$, for any $k \geq 4$, $a_k = \sum_{i=1}^{3} c_i a_i$, where $\{c_i\}_{i=1}^3$ are uniquely determined by $c_i = \langle a_k, a_i \rangle$. Then, $g a_k = g\left(\sum_{i=1}^{3} c_i a_i\right) = \sum_{i=1}^{3} c_i (g a_i) = \sum_{i=1}^{3} c_i b_i = b_k$.

Now, without loss of generality, we can conclude that if $\zeta(A) = \zeta(B)$, then there exists $g \in E(3)$ such that $gA = B$. As we have an injective map, our method is naturally at least as expressive as the GWL test for $E(3)$ isomorphism. As a result, our method surpasses all existing 3D GNNs in terms of distinguishing non-isomorphic point clouds.

$\square$

## A.2 Geometric Weisfeiler-Leman (GWL) Test

For the Geometric Weisfeiler-Leman (GWL) test, consider a graph $\mathcal{G}$ with its set of vertices represented as $\mathcal{V}(\mathcal{G})$ and its set of edges as $\mathcal{E}(\mathcal{G})$. A vertex in graph $\mathcal{G}$ is denoted by $i$, and $\mathcal{N}_i$ signifies the set of vertices adjacent to $i$. The color of vertex $i$ at iteration $t$ is given by $c_i^{(t)}$, and the geometric object for vertex $i$ at iteration $t$ is represented by $g_i^{(t)}$.

The procedure for the GWL test is as follows:

1. **Initialization**: Each vertex $i$ is assigned an initial color $c_i^{(0)}$ and a geometric object $g_i^{(0)}$, typically based on its local property or geometric attributes.

2. **Iterative Aggregation**: For each iteration $t \geq 1$, the geometric object of each vertex $i$ is updated to aggregate geometric information from its $t$-hop neighborhood, represented as $g_i^{(t)}$, which includes the colors and geometric objects from the previous iteration of vertex $i$ and its neighbors.

3. **Color Update**: The color of each vertex $i$ at iteration $t$ is computed by aggregating the geometric information around vertex $i$ using a $\mathfrak{G}$-orbit injective and $\mathfrak{G}$-invariant function, denoted by I-HASH, i.e., $c_i^{(t)} := \mathrm{I}^{-\mathrm{HASH}^{(t)}}\left(g_i^{(t)}\right)$.

4. **Termination**: The procedure terminates when colors do not change from the previous iteration or a predetermined maximum number of iterations is reached.

5. **Graph Comparison**: Finally, two geometric graphs $\mathcal{G}$ and $\mathcal{H}$ are geometrically non-isomorphic if there exists some iteration $t$ for which the sets of colors of their vertices are not equal, i.e., $\left\{\left\{c_i^{(t)} \mid i \in \mathcal{V}(\mathcal{G})\right\}\right\} \neq \left\{\left\{c_i^{(t)} \mid i \in \mathcal{V}(\mathcal{H})\right\}\right\}$.

## A.3 Statistical Moments

The equations that we used for calculating four moments are as follows.

The **mean**, often referred to as the average, represents the sum of all data points divided by the number of data points and is given by

$$\text{Mean} = \frac{\sum_{i=1}^{n} x_i}{n}. \tag{8}$$

**Variance** measures the spread or dispersion of a dataset and is defined as

$$\text{Variance} = \frac{\sum_{i=1}^{n}(x_i - \text{Mean})^2}{n-1}. \tag{9}$$

**Skewness** gauges the asymmetry of a dataset's distribution. Here we sightly change its definition to be positive for convenience as

$$\text{Skewness} = \frac{\sum_{i=1}^{n}|x_i - \text{Mean}|^3/n}{\{\sum_{i=1}^{n}(x_i - \text{Mean})^2/(n-1)\}^{3/2}}. \tag{10}$$

**Kurtosis** assesses the "tailedness" of a dataset's distribution as

$$\text{Kurtosis} = \frac{\sum_{i=1}^{n}(x_i - \text{Mean})^4/n}{\{\sum_{i=1}^{n}(x_i - \text{Mean})^2/(n-1)\}^2}. \tag{11}$$

### A.4 Schematic Diagram of our Framework

A schematic diagram of our active sampling framework is depicted in Fig. 6. We are given a labeled training set $L$, an unlabeled set $U$ and a query budget $k$ for each active learning iteration. The SphereNet model is first trained on the labeled set $L$. In the second step, the trained model is applied on the unlabeled set to compute a prediction uncertainty of each unlabeled molecule, which is used to populate the uncertainty vector $r$; the diversity matrix $D$ is also computed in this step where $D(i, j)$ is the diversity between unlabeled molecules $x_i$ and $x_j$. Next, the QP problem is solved to select $k$ unlabeled molecules for annotation. These molecules are removed from the unlabeled set $U$ and appended to the labeled set $L$. The active sampling process is continued iteratively until some stopping criterion is satisfied (taken as 7 iterations in our work).

Note that, computing the diversity matrix $D$ in Step 3 needs to be executed just once for the whole process. Once we have the initial $D$, as more and more samples are queried through AL, we keep deleting the corresponding rows and columns from $D$ to derive the updated matrix.

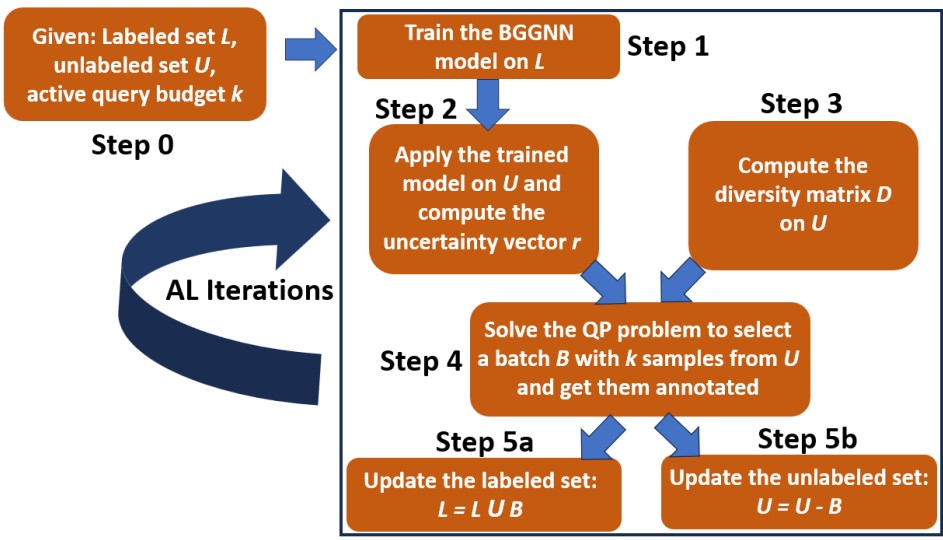

Figure 6: Schematic diagram of the proposed active learning framework.

### A.5 Related Work for Active Learning

Active Learning (AL) is a well-researched problem in the machine learning community [Settles, 2009]. Uncertainty sampling is an important strategy for AL, where unlabeled samples with the highest prediction uncertainties are queried for annotation. Several techniques have been explored to compute the uncertainty, such as Shannon's entropy [Guo and Schuurmans, 2007, Li and Guo, 2013], the distance of a sample from the separating hyperplane for SVM classifiers [Tong and Koller, 2001], the disagreement among a committee of classifiers regarding the label of a sample [Freund et al.,

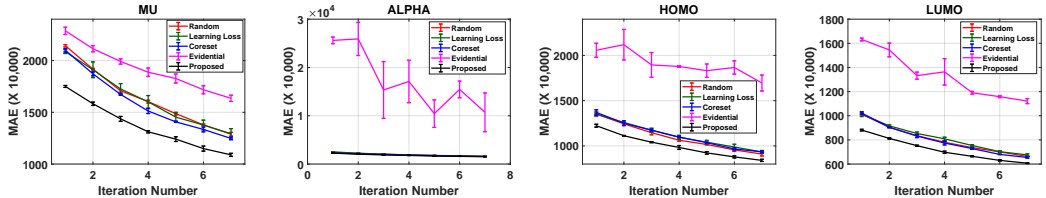

Figure 7: Active learning performance results. The graphs show the mean (averaged over 3 runs) and the errorbars for all the methods. Best viewed in color.

1997, Gilad-Bachrach et al., 2005], among others [Hoi et al., 2006, HOI et al., 2008, Guo and Greiner, 2007, Freytag et al., 2014]. With the advent of deep learning, Deep AL has attracted significant research attention [Hino, 2020, Ren et al., 2021], Entropy-based methods are developed as well [Wang and Shang, 2014, Ranganathan et al., 2017]. Yoo and Kweon [2019] cascaded a task-agnostic loss learning module that queries samples with the highest predicted loss values. Huang et al. [2021] proposed a strategy based on temporal output discrepancy. Techniques based on adversarial training have also been explored [Sinha et al., 2019, Mayer and Timofte, 2020, Zhang et al., 2020, Zhu and Bento, 2017]. Bayesian neural networks (BNNs) [Lampinen and Vehtari, 2001, Titterington, 2004, Goan and Fookes, 2020] and deep model ensemble [Lakshminarayanan et al., 2017, Huang et al., 2017] generally achieve superior performance but may induce excessive computational cost.

Diversity/representativeness based AL sampling has also been exploited. A core-set sampling technique proposed by **?** queries a batch of samples such that a model trained on the queried subset is competitive for the remaining data samples. Diversity sampling has also been exploited in the context of Bayesian neural networks [Kirsch et al., 2019]. Buchert et al. [2023] uses diversity sampling, together with self-supervised representation learning to select an informative seed set for AL. Combinations of uncertainty/diversity/representativeness-based criteria have also been used as query functions in AL research [Chakraborty et al., 2015, Wu et al., 2022, Ash et al., 2020].

### A.6  Results with the Evidential Uncertainty Baseline

The active learning performance results on the four properties studied (***mu, alpha, homo, and lumo***) are depicted in Fig. 7. As mentioned in Sec. 4.2, we note that *Evidential Uncertainty* depicts significantly high error values than the other methods, for all the four properties.

### A.7  Performance using the DimeNet$^{++}$ Backbone

The objective of this experiment is to study the performance of our framework with DimeNet$^{++}$ [Gasteiger et al., 2020a] as the backbone model of our active learning approach. We use the ***mu and lumo*** properties from the QM9 dataset in this experiment. We use the same experimental setup as detailed in Section 4.1 of the paper. The results are depicted in Figure 8. The proposed framework consistently outperforms all the baselines at each AL iteration across both the datasets.

The results of the statistical tests of significance are reported in Table 3. Each entry in the table denotes the p-value of the paired t-test between our method and the corresponding baseline (denoted in the columns) for the property studied (denoted in the rows). We note that the performance improvement achieved by our method is statistically significant ($p < 0.05$) compared to all the baselines for both the properties. These results corroborate the robustness of our framework to the underlying GNN backbone.

Table 3: The table shows the p-values obtained using paired t-test between the results our method against each of the baselines for the ***mu and lumo*** properties, using Dimenet$^{++}$ backbone.

| Properties | Baselines | | |
|---|---|---|---|
| | Random | Learning Loss | Coreset |
| ***mu*** | $1.56 \times 10^{-6}$ | $1.87 \times 10^{-4}$ | $1.25 \times 10^{-4}$ |
| ***lumo*** | $8.55 \times 10^{-4}$ | $4.24 \times 10^{-4}$ | $1.16 \times 10^{-2}$ |

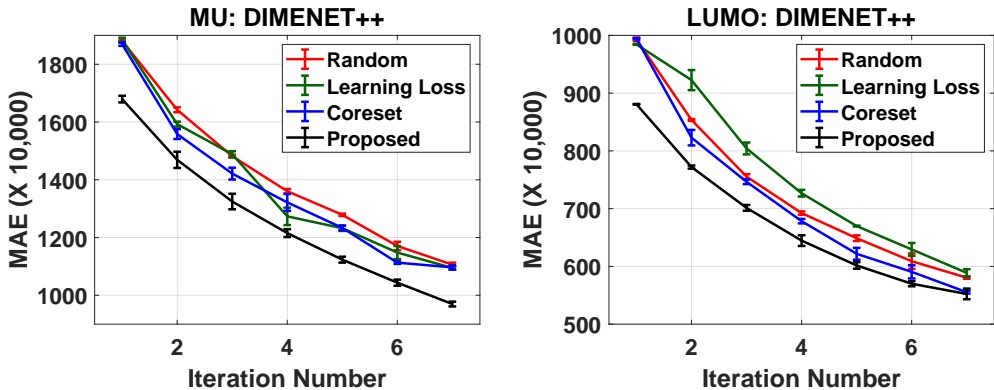

Figure 8: Study of our framework using the DimeNet$^{++}$ backbone. The graphs show the mean (averaged over 3 runs) and the errorbars for all the methods. Best viewed in color.

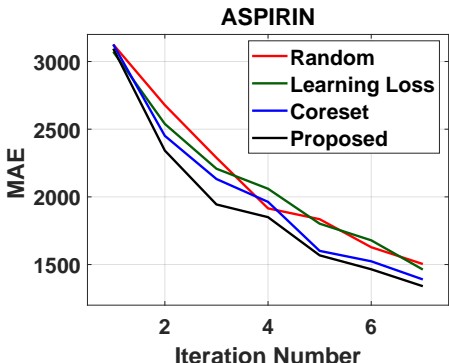

Figure 9: Study of our framework on Aspirin molecules using the SphereNet backbone. Best viewed in color.

### A.8   Generalization: Performance on the MD17 Dataset

The research of 3D molecular learning is new, and there are only a few reliable benchmark datasets for 3D molecules (containing atom types as well as XYZ coordinates for all atoms for each molecule). To test the generalization ability of our proposed method, we study the performance of our framework on the MD17 dataset [Chmiela et al., 2017a]. QM9 consists of molecules in equilibrium, while MD17 contains several thermalized (*i.e.*, non-equilibrium, slightly moving) molecular systems. Additionally, QM9 contains various quantum properties for molecules, like the important *homo* and *lomo* orbitals. MD17 is for dynamic system simulation, thus it contains labels for both the energy and atomic forces. In summary, we test our methods on molecule systems in both equilibrium and non-equilibrium, covering various quantum properties and molecular dynamics tasks.

We used 300 samples as the initial training set, 700 samples as the unlabeled set and $1,000$ test samples; we used 100 as the batch size and conducted 7 iterations of active learning. For this dataset, we train the network for 500 epochs. The results on *Aspirin* molecules using the SphereNet as the backbone of our GNN are depicted in Fig. 9. Our framework once again depicts promising performance and attains the lowest MAE values across all the AL iterations compared to all the baselines. These results further demonstrate the promise and potential of our method for scientific applications.

### A.9   Statistical Tests of Significance for the Query Budget Experiment

Table 4 reports the results of the statistical tests of significance for the study of query budget (presented in Sec. 4.3). Each entry in the table denotes the p-value of the paired t-test between our method

Table 4: The table shows the p-values obtained using paired t-test between the results our method against each of the baselines for the ***mu*** property for query budgets $1,000$, $1,500$ and $2,000$.

| Budget | Baselines | | | |
|---|---|---|---|---|
| | Random | Learning Loss | Coreset | Evidential |
| ***1000*** | $7.58 \times 10^{-6}$ | $1.05 \times 10^{-5}$ | $5.32 \times 10^{-5}$ | $2.46 \times 10^{-10}$ |
| ***1500*** | $7.54 \times 10^{-6}$ | $5.09 \times 10^{-5}$ | $1.51 \times 10^{-4}$ | $2.19 \times 10^{-7}$ |
| ***2000*** | $7.90 \times 10^{-5}$ | $1.74 \times 10^{-5}$ | $1.94 \times 10^{-4}$ | $1.77 \times 10^{-8}$ |

and the corresponding baseline (denoted in the columns) for the query budget (denoted in the rows) for the ***mu*** property. From the table, we note that the improvement in performance achieved by our method is statistically significant ($p < 0.05$) compared to all the baselines, consistently for all the query budgets.

### A.10    Addtional Ablation Studies

**The Individual Impact of Diversity and Uncertainty Components.** In Fig. 10, we present the result on the individual impact of the diversity and uncertainty components. Our proposed method outperforms the individual use of diversity or uncertainty alone. The key to this outperformance lies in our method's dual focus on both geometric importance and chemical contexts. Moreover, it can be observed that the diversity component alone shows strong performance; it is only slightly less effective than our method because it captures the geometries of molecules, which are fundamental in distinguishing different molecules with different properties. On top of this, we also conduct statistical tests to conclude that the improvement of our method is significant compared to only diversity or only uncertainty in Table 5.

Table 5: The table shows the p-values obtained using paired t-test between the result of our method against uncertainty only and diversity only components in ablation study for ***mu*** and ***lumo*** prediction.

| Properties | Components | |
|---|---|---|
| | Uncertainty Only | Diversity Only |
| ***mu*** | $1.24 \times 10^{-5}$ | $1.82 \times 10^{-4}$ |
| ***lumo*** | $7.46 \times 10^{-6}$ | $2.26 \times 10^{-4}$ |

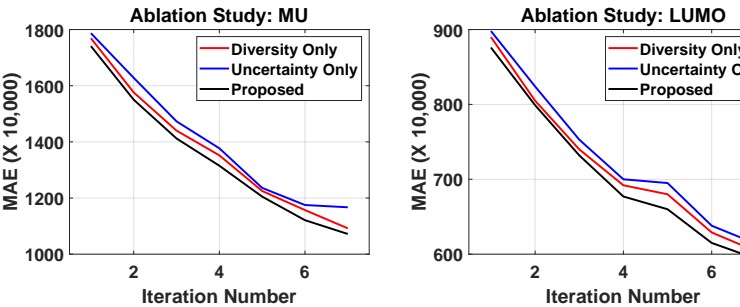

Figure 10: Ablation study results studying the individual impact of uncertainty and diversity on the ***mu*** and ***lumo*** properties with SphereNet. Best viewed in color.

**A Comparison between Our Diversity Component and SOAP.** We include a comparison between our method and one that uses a well-known geometric descriptor in chemistry, the SOAP descriptor [Bartók et al., 2013]. SOAP produces descriptors that characterize local atomic environments using spherical harmonics and radial basis functions. It incorporates both geometric information and elemental (species) details. Table 6 presents the results on the QM9 dataset for two important properties, *mu* and *lumo*. These results demonstrate that our diversity component clearly outperforms SOAP. Consequently, our overall method, which combines both diversity and uncertainty, also

surpasses the performance of the SOAP descriptors. The p-values in Table 7 further illustrate that our proposed method significantly improves the selection strategy compared to SOAP. This improvement can be attributed to the more localized nature of the SOAP descriptors and the inability to maintain permutation invariance.

Table 6: The table shows a comparison between our proposed descriptor and the SOAP descriptor for the properties ***mu*** and ***lumo***, using SphereNet as the backbone.

| Iteration | *mu* | | | | *lumo* | | | |
|---|---|---|---|---|---|---|---|---|
| | SOAP D | Our D | SOAP B | Our B | SOAP D | Our D | SOAP B | Our B |
| 1 | 2154 | **1769** | 2057 | **1741** | 1016 | **890** | 1013 | **876** |
| 2 | 1901 | **1576** | 1877 | **1550** | 921 | **805** | 902 | **799** |
| 3 | 1732 | **1440** | 1721 | **1412** | 839 | **740** | 838 | **732** |
| 4 | 1701 | **1352** | 1539 | **1315** | 797 | **692** | 791 | **677** |
| 5 | 1587 | **1225** | 1456 | **1205** | 759 | **680** | 744 | **660** |
| 6 | 1414 | **1157** | 1345 | **1121** | 699 | **629** | 711 | **615** |
| 7 | 1322 | **1092** | 1280 | **1072** | 667 | **604** | 681 | **594** |

Abbreviations: D means using diversity Only; B means using both uncertainty + diversity.
The results from the method with superior performance are highlighted in bold.

Table 7: The table shows the p-values obtained from a paired t-test comparing the results of our method against those of SOAP for the properties ***mu*** and *lumo*, using SphereNet as the backbone.

| | *mu* | *lumo* |
|---|---|---|
| *p-value* | $4.20 \times 10^{-6}$ | $2.51 \times 10^{-6}$ |

**A Comparison between our method and BatchBALD.** We investigate the impact of our quadratic programming formulation compared to the greedy, clustering-based Bayesian uncertainty baseline, BatchBALD [Kirsch et al., 2019], which selects a diverse batch of samples by maximizing mutual information. Our method provides a more structured approach to uncertainty, particularly tailored for 3D molecular data. The results, presented in Fig. 11, demonstrate the effectiveness of our approach. Additionally, we conducted statistical tests, as shown in Table 8, which confirm that the improvement of our method over the baselines is statistically significant. This outperformance can be attributed to the components specifically designed for 3D molecular graphs.

Table 8: The p-values obtained using paired t-test between the results our method against each of the baselines for all the properties studied. *Here, L. Loss refers to Learning Loss.*

| Properties | Baselines | | | | |
|---|---|---|---|---|---|
| | Random | L. Loss | Coreset | Evidential | BatchBALD |
| ***mu*** | $7.54 \times 10^{-6}$ | $5.09 \times 10^{-5}$ | $1.51 \times 10^{-4}$ | $2.19 \times 10^{-7}$ | $7.20 \times 10^{-5}$ |
| ***alpha*** | $1.06 \times 10^{-5}$ | $8.14 \times 10^{-4}$ | $4.27 \times 10^{-5}$ | $2.72 \times 10^{-4}$ | $4.86 \times 10^{-6}$ |
| ***homo*** | $2.26 \times 10^{-5}$ | $8.36 \times 10^{-7}$ | $4.23 \times 10^{-6}$ | $1.71 \times 10^{-8}$ | $1.54 \times 10^{-4}$ |
| ***lumo*** | $4.48 \times 10^{-5}$ | $1.25 \times 10^{-5}$ | $3.12 \times 10^{-4}$ | $2.39 \times 10^{-6}$ | $3.96 \times 10^{-5}$ |

## A.11   Licenses for Existing Assets

We list all the licenses for existing assets in Table 9.

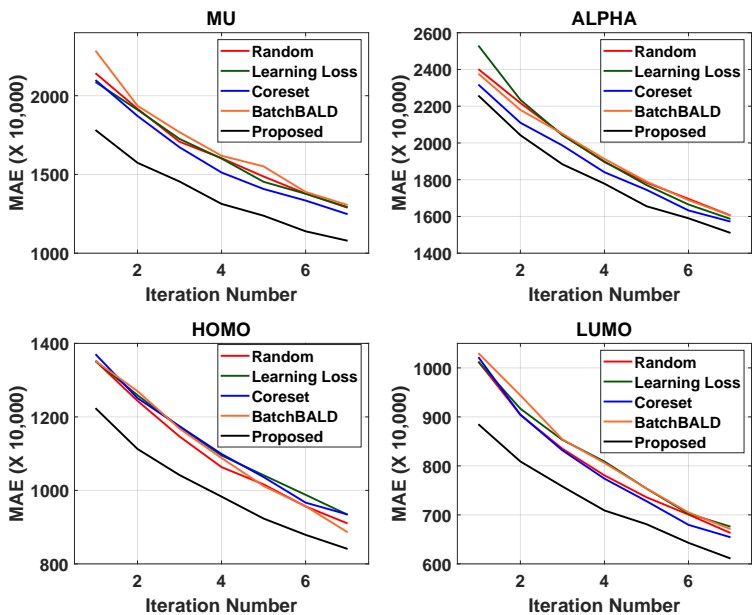

Figure 11: Active learning performance results with SphereNet on QM9 Dataset. Best viewed in color.

Table 9: Assets, Licenses, and Descriptions

| Asset | License | Description |
|---|---|---|
| SphereNet [Liu et al., 2022] | GNU General Public License v3.0 | GNN Model |
| DimeNet$^{++}$ [Gasteiger et al., 2020a] | Hippocratic License v2.1 | GNN Model |
| QM9 [Ramakrishnan et al., 2014] | CC BY-NC-SA 4.0 International License | Benchmark Dataset |
| MD17 [Ramakrishnan et al., 2014] | CC BY-NC-SA 4.0 International License | Benchmark Dataset |
| Coreset [Sener and Savarese, 2018] | MIT License | Active Learning Scheme |
| Learning Loss [Yoo and Kweon, 2019] | N/A | Active Learning Scheme |
| Evidential Uncertainty [Beluch et al., 2018, Amini et al., 2020] | N/A, Apache-2.0 License | Active Learning Scheme |

