# OpenReview forum: "Empowering Active Learning for 3D Molecular Graphs with Geometric Graph Isomorphism"
_NeurIPS.cc/2024/Conference — NeurIPS 2024 poster_

### Official Review · Reviewer_Abqb · 2024-06-24

**Soundness:** 3
**Presentation:** 2
**Contribution:** 3
**Rating:** 5
**Confidence:** 3

**Summary:**

This paper proposes a principled AL paradigm to alleviate the annotation hurdle of 3D molecular graphs. It introduces a novel diversity component for 3D molecular graphs, which is provably at least as expressive as the GWL test. Furthermore, the authors develop an effective and efficient pipeline to compute uncertainties for 3D molecular graphs rooted in Bayesian inference.

**Strengths:**

+ The authors develop an effective method for computing diversity among different 3D molecular graphs and introduce a Bayesian Geometric Graph Neural Network (BGGNN). The BGGNN takes a 3D graph as input and produces the desired properties along with uncertainty values.
+ The motivation of the paper is clear, and it includes well-defined theoretical proofs.

**Weaknesses:**

+ The paper lacks some necessary explanations, such as those for "USR" and "GSL," making it difficult to understand.
+ Experiments were conducted only on the QM9 and MD17 datasets, so the effectiveness and efficiency on larger datasets remain unknown.

**Questions:**

+ The authors formulate a criterion based on uncertainty and diversity. How do these two factors respectively impact performance?

**Limitations:**

Please see the weaknesses part.

---

> ### Author Rebuttal · Authors · 2024-08-03
>
> > W1: lack of necessary explanations
>
> Thank you for your comments.
>
> For "USR", the primary concept involves using statistical moments to approximate the geometry of the molecules, capturing essential features of their shapes. Detailed explanations of this method are provided in Section 2.1.3 (lines 189 to 214). If you feel that additional background information is needed, please let us know the specific points where more details would be beneficial so that we can improve our explanation.
>
> We do not have the abbreviation "GSL" in the paper. If you mean "GWL," we present the high-level ideas of the GWL test from lines 159 to 164. The key message we want to convey is, as stated in the paper, "the GWL test imposes an upper bound on the expressive power of 3D GNNs." We also provide the details of this test in Appendix A.2.
>
> We believe that the high-level explanations highlighted in the main paper are sufficient to understand our claims. We also provide detailed explanations in Appendix. Therefore, we kindly request not to view this as a weakness of our work.
> However, we will definitely take your suggestions and further clarify the context.
>
> > W2: experiments on larger datasets
>
> While our experiments were conducted exclusively on the QM9 and MD17 datasets, this choice was made because our primary focus is on small molecules and to validate our method on **well-established** and **most commonly used** benchmarks.
> QM9 and MD17 are necessary benchmark datasets in the field of 3D molecular learning. Almost all representative works in the field use them for evaluation [1-4].
>
> Other reliable benchmark datasets for 3D scientific data would go beyond molecules, like the Materials Project dataset for crystal materials and the Fold Dataset for proteins. These data contain special structures (e.g., periodic structures for crystals), and thus they are out of the scope of this work. We will explore the potential of our active learning methods on these macromolecules as future work.
>
> If you have concerns about complexity, we can vectorize the diversity computation implementation for GPUs for faster computations. We have updated the results comparing selection times (please see Table 3 of the attached PDF) and included additional discussions in the global response (item 3), to reflect our new vectorized implementation. It can be seen that our sampling approach is highly effective and efficient. Note that the Coreset approach is an important diversity-based baseline, and our sampling approach is much more efficient than it (127 vs 64.9 minutes). The Random approach does not involve any sophisticated sampling strategy, and our method is just slightly more expensive than it (53 vs 64.9 minutes). Thus, we believe our sampling approach is readily applicable to larger datasets.
>
>
> > Q1: individual impact of uncertainty and diversity
>
> We have shown in the ablation study (Sec. 4.4) that our proposed diversity metric alone is highly effective. We can statistically conclude that with only the proposed diversity metric, we outperform all other baselines. Additionally, the uncertainty metric clearly further improves our method as all the results suggest. **In summary, our ablation study (Sec. 4.4) demonstrates that our proposed diversity metric is highly effective on its own. Statistically, we outperform all other baselines using only this metric. Furthermore, incorporating the uncertainty metric enhances our method even more.**
>
> Additionally, **we have presented new ablation results in the global response** (Figure 2 and Table 2), please see the details there. We compared our full method (combining uncertainty and diversity) against methods using only uncertainty or only diversity. The results indicate that our diversity component is highly effective in selecting informative samples for model training. Moreover, the performance improvement achieved by our method compared to using only uncertainty or only diversity-based selection is statistically significant ($p < 0.001$), as shown in Table 2 of the PDF file in the global rebuttal. Incorporating uncertainty further refines our method by integrating additional chemical contexts, such as atom types. **This clearly demonstrates that our approach is superior, as it not only distinguishes different geometries with precision but also accounts for uncertainties in chemical contexts.**
>
> Reference
>
> [1] K.S., et al. SchNet: A continuous-filter convolutional neural network for modeling quantum interactions. NIPS'17
>
> [2] J.K., et al. Directional message passing for molecular graphs. ICLR'20
>
> [3] Y. L., et al. Spherical message passing for 3D molecular graphs. ICLR'22
>
> [4] J. G., et al. GemNet: Universal Directional Graph Neural Networks for Molecules. NeurIPS'21

---

> > ### Comment · Reviewer_Abqb · 2024-08-13
> > **Thank you.**
> >
> > Thank you for your thoughtful response. My concern has been addressed. I will increase the score from 4 to 5.

---

> ### Author Response · Authors · 2024-08-13
> **The reviewer-author discussion stage is ending soon**
>
> Dear Reviewer Abqb,
>
> Thank you for your comments! As **the reviewer-author discussion stage is ending in less than 30 hours**, we kindly remind you that we have provided a detailed rebuttal (both the individual rebuttal and global rebuttal) to address your concerns. We hope our clarifications and additional experiments (in the global rebuttal PDF file) have resolved the concerns to your satisfaction.
>
>  If you believe your concerns have been adequately addressed, we kindly ask you to reconsider your scores. We are more than willing to discuss them in detail if you have additional questions.
>
> Sincerely,
>
> Authors

---

> ### Author Response · Authors · 2024-08-13
>
> Dear Reviewer Abqb,
>
> Thank you for your response and for increasing your score.
> We are happy to know that your concerns have been addressed.
>
> Sincerely,
>
> Authors

---

### Official Review · Reviewer_JSdU · 2024-07-03

**Soundness:** 3
**Presentation:** 3
**Contribution:** 3
**Rating:** 7
**Confidence:** 2

**Summary:**

This paper introduces a principled active learning (AL) paradigm tailored for molecular learning. The proposed AL approach aims to alleviate the hurdle of human annotation by automatically querying labels for the most informative samples. The authors treat molecules as 3D molecular graphs and they introduce a set of new 3D graph isometries for 3D graph isomorphism analysis, which are shown to be as expressive as the Geometric Weisfeiler-Lehman (GWL) test. To ensure the selection of samples with maximal uncertainties, the authors design a Bayesian geometric graph neural network specifically for 3D molecular graphs. Active sampling is formulated as a quadratic programming (QP) problem integrating these components. Experimental results demonstrate the effectiveness of the proposed AL paradigm, highlighting the advantages of the diversity and uncertainty methods introduced.

**Strengths:**

1. **Innovative Isometries for Graph Discrimination**: The introduction of a new set of 3D graph isometries is a significant contribution, demonstrating important efficacy in distinguishing between different graphs. This advance enhances the expressiveness and applicability of the proposed method.

2. **Theoretical Support and Clarity**: The methodology is explained with theoretical support

3. **Comprehensive Experimental Evaluation**: The experiments clearly highlight the method's improvements over existing baselines. The inclusion of statistical analysis further strengthens the validity of the results, providing evidence of the method's effectiveness.

**Weaknesses:**

1. **Overly Strong Claims**: Some of the claims made in the paper are quite strong and require more careful justification. For detailed points of concern, please refer to the Questions section. Providing additional evidence or more nuanced discussions could strengthen these claims.

2. **Need for More Detailed Experimental Insights**: While the experimental section is comprehensive, it could benefit from additional insights and details. Specific experiments would be clearer and more informative with further explanation. Providing more context and interpretation of the results would enhance the reader's understanding and the overall impact of the findings.

**Questions:**

1. **Clarification Needed on Line 175**: The sentence starting in line 175 needs to be more carefully justified. Specifically, how can you assert that SchNet is upper bounded by distance isometry but not by triangular and cross-angular isometries? Please provide a more detailed explanation or supporting evidence for this claim.

2. **Generalization in Section 4.2**: It is not clear how the experiment in Section 4.2 demonstrates generalization. Could you provide more information and insight on the choice of graphs used in this section? Additionally, explaining the rationale behind the experimental setup and how it supports your conclusions would be helpful.

3. **Complete Results for Section 4.4**: Please report all the results from Section 4.4.

**Limitations:**

Limitations addressed

---

> ### Author Rebuttal · Authors · 2024-08-03
>
> > W1: Overly Strong Claims
>
> Thank you for your comments. We will use more precise language and include additional discussions in the paper to clarify our claims. We will address your concerns in the Questions section.
>
> > W2: Need for More Detailed Experimental Insights
>
> In the current version of our paper, we have provided the necessary information for the overall experimental setup, as well as detailed analyses for each individual experiment. Even so, we agree with you that some rationales and high-level insights are missing, especially for readers outside this field to easily understand the paper. As we cannot revise the papers for now, we summarize the key rationales and insights in this rebuttal; **we will integrate them in our next version.**
>
> - 4.1 Experiment setup: We choose different types of molecular graphs for various tasks to demonstrate the generalizability of our methods.  We test our methods on molecular systems in both equilibrium and non-equilibrium states, covering various quantum properties and molecular dynamics tasks. This will be further detailed for your Question 2 below.
>
> - 4.2 The performance comparison of our method with the baselines: This shows that our active sampling approach is significantly better than mainstream active learning methods. This further means that, given a budget to select more informative samples for wet-lab annotation (usually expensive for molecules), our method can select the most informative subset, thereby significantly improving the learning efficacy and efficiency.
>
> - 4.3 An examination of the query budget study: This shows that for various annotation budgets, our method is consistently better than baselines, which indicates that the superior capability of our sampling approach is robust in practice.
>
> - 4.4 An ablation study focusing separately on the diversity and uncertainty components: This indicates that both the proposed sampling approaches are novel and effective. Combining them considers both the geometry and chemical contexts in 3D molecules, and thus achieving the best active learning performance.
>
> - 4.5 An evaluation of the computation time: This shows our method is not only effective but also efficient. Our sampling approach has similar efficiency to the Random sampling, and is much more efficient than the important diversity-based active learning baseline Coreset.
> This further indicates our approach is readily applicable to macromolecules like materials and proteins, which will be conducted as our future work.
>
>
>
> > Q1: Clarification Needed on Line 175
>
> To clarify, we mean that SchNet has at most the same expressive power in distinguishing different geometric structures as our method. Since both methods rely solely on distance information, they can distinguish structures uniquely defined by their distances alone. Our method is deterministic, meaning that, with the same distance information, **a perfectly trained SchNet can be at most as powerful as our method**. In practice, it is almost impossible for a SchNet to be perfectly trained.
>
> > Q2: Generalization in Section 4.2
>
> The research of 3D molecular learning is new, and there are only a few reliable benchmark datasets for 3D molecules (containing atom types as well as XYZ coordinates for all atoms for each molecule). We choose our datasets based on the following two criteria:
>
> QM9 consists of molecules in equilibrium, while MD17 contains several thermalized (i.e. non-equilibrium, slightly moving) molecular systems. Additionally, QM9 contains various quantum properties for molecules, like the important HOMO and LOMO orbitals. MD17 is for dynamic system simulation, thus it contains labels for both the energy and atomic forces. In summary, we test our methods on molecule systems in both equilibrium and non-equilibrium, covering various quantum properties and molecular dynamics tasks.
>
> QM9 and MD17 are necessary benchmark datasets in the field of 3D molecular learning. Almost all representative works in the field use them for evaluation [1-4]. We follow this conventional setup in our work. Besides QM9 and MD17, other reliable benchmark datasets for 3D scientific data would go beyond molecules, like the Materials Project dataset for crystal materials and the Fold Dataset for proteins. These data contain special structures (like periodic structures for crystals), and thus they are out of the scope of this work. We will explore the potential of our active learning methods on these data as future work.
>
> We will include these discussions in the next version of our paper.
>
> > Q3: Complete Results for Section 4.4
>
> We have included additional results from the ablation study, which can be found in the global response (please refer to Figure 2 and Table 2 of the attached PDF file ). We've also included more discussions in item 2 of the global rebuttal. Basically, the results show that both the uncertainty-only and diversity-only approaches outperform all baselines. Additionally, we show both diversity and uncertainty components significantly contribute to the overall performance. We did not include other baselines in this plot to avoid overwhelming it, but if by "report all the results from Section 4.4", you mean including all the baselines in this plot as well, we will certainly do so in the next version of the paper.
>
> Reference
>
> [1] K.S., et al.  SchNet: A continuous-filter convolutional neural network for modeling quantum interactions. NIPS'17
>
> [2] J.K., et al. Directional message passing for molecular graphs. ICLR'20
>
> [3] Y. L., et al. Spherical message passing for 3D molecular graphs. ICLR'22
>
> [4] J. G., et al. GemNet: Universal Directional Graph Neural Networks for Molecules. NeurIPS'21

---

> > ### Comment · Reviewer_JSdU · 2024-08-12
> >
> > I appreciate the authors' responses. Including all the baselines from the experiments in Section 4.4 would enhance the completeness of the paper, and I strongly encourage adding them to the appendix. Given that my concerns have been satisfactorily addressed, I am willing to increase my score.

---

> ### Author Response · Authors · 2024-08-12
>
> Dear Reviewer JSdU,
>
> Thank you so much for your constructive comments, which are very helpful in improving the clarity of our paper.
> Also thanks for acknowledging that we've addressed your concerns and raising your score.
>
> As we cannot revise the paper for now, we'll definitely include these new results either in the main paper or in the Appendix in the next version.
>
> Sincerely,
>
> Authors

---

### Official Review · Reviewer_UZqH · 2024-07-09

**Soundness:** 3
**Presentation:** 3
**Contribution:** 3
**Rating:** 6
**Confidence:** 4

**Summary:**

The paper proposes an active learning scheme for molecular property prediction using uncertainty estimates from Dropout Monte Carlo and diversity metrics. In each active learning iteration, molecules are selected by maximizing uncertainty and diversity in the batch by solving a quadratic programming problem. The authors propose a 48-dimensional vector that encodes statistical moments of reference distances, angles and cross-angles for 4 reference points. Experiments on QM9 and MD17 datasets are performed to benchmark the effectiveness of the proposed method against several baselines.

**Strengths:**

1.	The proposed 48-dimensional vector that encodes statistical moments of reference distances, angles and cross-angles for 4 reference points is a promising and novel metric to quantify geometric diversity of 3D molecular structures.
2.	The proposed quadratic programming formulation of the molecule selection step allows to trade-off uncertainty and diversity of selected conformations.
3.	The performed experiments demonstrate the usefulness of the proposed diversity metric and clearly outperforms the random selection baseline and other, non-Bayesian-uncertainty-based active learning approaches.

**Weaknesses:**

1.	Many of the claims in the paper are too bold, neglecting several important related works in the field:
     - There are several active learning applications for 3D GNNs already in the literature, especially in the neural network potential literature:  https://www.nature.com/articles/s41467-021-21376-0 ,   https://www.nature.com/articles/s41524-023-01104-6
     - In the review of alternative 3D structural descriptors, the most common methods such as SOAP ( https://journals.aps.org/prb/abstract/10.1103/PhysRevB.87.184115  ), or ACE ( https://journals.aps.org/prb/abstract/10.1103/PhysRevB.99.014104 ) are neglected.
     - Unlike claimed by the authors, the proposed Dropout Monte Carlo scheme for active learning in molecular property prediction with 3D GNNs is not novel, but has been proposed previously: https://www.nature.com/articles/s41524-024-01277-8

2.	The considered baselines might not be the most relevant to the method and do not allow to attribute the origin of the empirical performance increase given that neither Coreset nor Learning loss uses a Bayesian uncertainty. BatchBALD would be a much more relevant baseline given that the same uncertainty estimates can be used. This would also allow to investigate the effect of the quadratic programming formulation against the greedy clustering-based approach of BarchBALD. It would also allow to investigate the benefit of the proposed diversity metric vector against, for example, classical descriptors such as SOAP. This would allow to clarify the underlying source of the outperformance: Is it the Bayesian UQ estimate, the quadratic programming formulation of the proposed novel structure descriptor vector?

3.	The quadratic scaling of the computational cost of the method is concerning. The pool sizes considered in the experimental section are very small (15,000) in the field of 3D molecular property prediction, where datasets can easily span several million conformations (PubChemQC, ANI-2x, GEOM). Given that there is already a computational overhead visible over competing AL methods, this difference will only increase for realistically-size datasets.

**Questions:**

Does the 48-dimensional vector contain any information about the atom-species involved in distances, angles and cross angles or is this metric purely geometry-based?

**Limitations:**

The authors discuss some limitations of the proposed approach, including the scaling to large datasets such as OC20. However, the scaling to large atom sizes could be discussed more as requiring further research given that it is non-obvious that the statistical moments measured at 4 reference points are sufficient to distinguish protein-size geometries or large crystal structures. Additionally, it should be stated more clearly that the proven power of the proposed descriptors exceeding the GWL test only holds before compressing the descriptors into the 48-dimensional vector.

---

> ### Author Rebuttal · Authors · 2024-08-03
>
> > W1: Many of the claims in the paper are too bold, neglecting several important related works in the field.
>
> Thank you for your comments. We will modify the language in the paper to clarify our intentions.
>
> The first two active learning (AL) papers do not specifically consider 3D geometry information and **differ from our paper/method**, which focuses on 3D molecules. The 3D geometry of molecules is crucial in determining molecular properties, but entangles unique challenges for designing AL schemes, which motivates our work.
> We will make sure to include them in the literature review.
>
> Regarding the descriptors, our method produces a 48-dim vector **describing the geometry of a molecule**. Our method is equivariant to roto-translations and **invariant to permutations**, as the statistical quantities do not change due to permutation. Meanwhile, SOAP produces vectors that **describe local atomic environments** using spherical harmonics and radial basis functions in an atom-wise manner. In general, they are equivariant to roto-translations but **not invariant to permutations**. ACE involves a systematic expansion that can describe various orders of interactions (e.g., two-body, three-body); however, it is **less of a conventional descriptor compared to our method and SOAP**. We will make sure to include these two papers in our discussions in the paper.
>
> Regarding the Monte Carlo scheme, this paper was published on May 3, 2024, which is only around 20 days before the submission deadline. We were not aware of this paper at the time. We would like to say that this is **concurrent work**, and we will discuss this paper in our next version.
>
> > W2: Bayesian baseline and SOAP descriptors
>
> We have included the results for BatchBALD in the global rebuttal (please refer to Figure 1 of the attached PDF). It can be clearly seen that **our method outperforms BatchBALD**. Moreover, we have performed statistical tests to confirm that **our improvement compared to BatchBALD is significant** with a $p$-value << $0.001$ (please refer to Table 1 of the attached PDF).
>
> In regards to the SOAP descriptors, as we discussed in W1, SOAP produces vectors that describe local atomic environments, whereas we aim to measure diversity based on global features. There are ways to concatenate and compress the SOAP descriptors into lower-dimensional global descriptors; however, given the short time frame of the rebuttal period, we cannot provide any results on this. We will definitely consider working on this in the future.
>
> > W3: computational cost
>
> First, an active sapling strategy will rely on the backbone 3D GNN models for downstream molecular learning. However, existing mainstream 3D GNNs have the complexity of $O(n^2)$ (DimeNet [1], SphereNet [2]) or even $O(n^3)$ (GemNet [3]). Hence, the proposed sampling approach with $O(n^2)$ does not increase the order of the overall complexity, and might not be the major computational overhead of the whole learning pipeline.
>
> As mentioned in Sec. 2.3 of the paper, we implemented a solution to execute the QP problem on the GPU (instead of the CPU) using the parallel implementation of the alternating direction method of multipliers, as detailed in [4]. Furthermore, we have vectorized our implementation and utilized the GPU to perform calculations to make the quadratic diversity matrix computation faster (find more details in item 3 the global rebuttal). Overall, our efficiency is slightly worse than that of the basic Random sampling.
>
> For larger datasets, we can also address scalability by first sub-sampling from the unlabeled pool using only the uncertainty criterion (which is linear and thus scalable) to select the most uncertain samples. We then apply the proposed active sampling criterion only to the selected subset. This strategy has been used in previous AL research with promising results [5]. We plan to explore this strategy on large-scale molecular datasets in the future.
>
> > Q1: 48-dim vector
>
> It is purely geometry-based. However, we want to emphasize that our framework contains two components for selecting important molecules: diversity and uncertainty. The diversity component focuses on the geometric perspective of view.
> The uncertainty part considers atom types, which are embedded into node features. By combining both uncertainty and diversity, **we take both the chemical contexts and geometric contexts into consideration**.
>
>
> > L1: scaling to large atom sizes with statistical moments and expressive power after compressing the descriptors
>
> Thank you for your comments, we will include such a discussion in the limitation section.
>
> USR is a well-known work for recognizing similar molecular shapes [6], where they use distance information only and employ the first three moments (mean, variance, skewness) to approximate the distribution of distances. The authors state in the paper (page 5, right column, 2nd paragraph) that "Such an approach is based on a theorem [7] from statistics, which proves that a distribution is completely determined by its moments."
>
> With this theorem in mind, we can reconstruct the distribution with high fidelity by increasing the number of translated moments as well as the order of computed moments. We found that **distance information alone cannot describe a complete isometry space**, so we study a complete isometry space by **considering angular isometries**, eventually resulting in a theoretically guaranteed solution for precise molecular diversity computing. With the proposed complete isometry space and sufficient statistical moments, our method can be at least as expressive as the GWL test. However, in practice, we compress into a lower-dimensional descriptor for efficiency.
>
> This also addresses the scaling issue. For protein-sized geometries or large crystal structures, one can choose to include more statistical moments to better distinguish different structures. As our work focuses on small molecules, we use four statistical moments.

---

> > ### Comment · Reviewer_UZqH · 2024-08-08
> >
> > > The first two active learning (AL) papers do not specifically consider 3D geometry information
> >
> > These works do train 3D GNNs via active learning. They do not perform any diversity-based selection, but the models and datasets are inherently 3D. When the authors claim that their model uses the atomic species via the uncertainty-metric, so do these works consider the 3D structure via the uncertainty metric.
> >
> > > SOAP descriptors
> >
> > It is unfortunate that there are no results on this given that the proposed 48-dim vector seems to be the biggest novelty of the paper, so a comparison to the clustering capabilities to standard approaches such as SOAP (acknowledging the fact that another step is required to obtain a global descriptor from the local descriptors) would be highly valuable.
> >
> > > existing mainstream 3D GNNs have the complexity of O(n^2)
> >
> > (Assuming n is the number of atoms in the system) Then these 3D GNN backbones have have complexities of O(d^p n), i.e. are linear in the number of atoms, times the average number of neighbors d (which is a constant that is depending on the cut_off and the density of the system, but independent of n). If the proposed method is indeed O(n^2), this might become problematic for larger systems in case the cost starts to dominate the cost of the backbone.
> >
> > > Q1: 48-dim vector: It is purely geometry-based
> >
> > This should be noted in the limitations given that 2 molecules with similar geometries, but different atom species tend to behave very differently, even though the 48-dim vector treats them very similarly.

---

> ### Author Response · Authors · 2024-08-07
> **References to Rebuttal**
>
> References
>
> [1] J.K., et al. Directional message passing for molecular graphs. ICLR'20
>
> [2] Y. L., et al. Spherical message passing for 3D molecular graphs. ICLR'22
>
> [3] J. G., et al. GemNet: Universal Directional Graph Neural Networks for Molecules. NeurIPS'21
>
> [4] M.G., et al. GPU acceleration of admm for large-scale quadratic programming. Journal of Parallel and Distributed Computing
>
> [5] S. C., et al. Active batch selection via convex relaxations with guaranteed solution bounds. TPAMI
>
> [6] P. J B., et al. Ultrafast shape recognition to search compound databases for similar molecular shapes. Journal of computational chemistry
>
> [7] Hall P. A distribution is completely determined by its translated moments. Zeitschrift für Wahrscheinlichkeitstheorie und Verwandte Gebiete.

---

> ### Author Response · Authors · 2024-08-10
> **Reply to Reviewer UZqH (Part I)**
>
> Thank you for your prompt reply. Your insights are invaluable to us.
>
> > These works do train 3D GNNs via active learning...
>
> First, we apologize for any confusion caused by our language. When we said that they do not specifically consider 3D geometry information, we meant that the active learning primarily pertains to 3D geometries, as in our diversity metric.
>
> The scopes of the mentioned two methods [1,2] are slightly different, but for the active selection part, both methods sample configurations in sub-regions where the machine learning models have maximal **uncertainty** about for quantum mechanics (e.g., DFT) calculations, to be added to training data. In performing similar tasks (e.g., in our MD17 Benzene experiments), their methods share some similarities with our uncertainty component. However, their methods specifically targets non-equilibrium state modeling, while our method is more general, applicable to molecular systems in both equilibrium (e.g., QM9 experiments) and non-equilibrium states, covering various quantum properties and molecular dynamics simulation tasks.
>
> Moreover, our approach incorporates both uncertainty and diversity. While 3D information is considered in active learning approaches such as [1, 2], it is not as central as in our diversity computations. Usually, the models (3D GNNs) play an important role in uncertainty quantification; since many node features, other than geometries, are fed as inputs to the model, uncertainty quantification may or may not prioritize 3D geometric information. Our diversity component is laser-focused on 3D geometric information. While pure uncertainty-based methods focus solely on the most uncertain samples, our method balances exploration and exploitation, leading to more comprehensive and reliable sample selection. Experiments in Sec. 4.4 in our paper and Fig. 2 in the global rebuttal PDF show that, our method consistently outperforms the individual use of uncertainty for all active learning iterations.
>
> We understand your point of view that these methods also consider 3D information, although not as specific as our method (especially the diversity component). We will make sure to include these references, discuss our focus in further detail, and modify our sentences to avoid bold claims, as you mentioned in your first reply.
>
> > It is unfortunate that there are no results on this given that the proposed 48-dim vector seems to be the biggest novelty of the paper, so a comparison to the clustering capabilities to standard approaches such as SOAP (acknowledging the fact that another step is required to obtain a global descriptor from the local descriptors) would be highly valuable.
>
> Unfortunately, we did not have enough time to complete the experiments during the rebuttal period. However, during this discussion period, we managed to perform the experiments using the SOAP descriptor and compare its clustering capabilities to our 48-dimensional vector. We first obtain the local SOAP descriptors and then aggregate them to obtain a global descriptor for a molecule. Then, we apply the same setting as mentioned in section 2.3 of the paper to obtain the results.
>
> We herein present the results. The reported values are Mean Absolute Error (MAE) values.
>
> *Abbreviations:*
> D= Diversity Only, B= Uncertainty + Diversity
>
> | Iteration |  |*mu* |  |  |*lumo*| |
> |----------|------|-|---|----|----|--|
> |   | SOAP D| Our D | Our B |&#124;   SOAP D| Our D | Our B |
> | 1 | 2154|  1769| 1741 |   &#124;   1016 |    890  | 876
> | 2 | 1901|  1576| 1550 |   &#124; 921 |    805  | 799
> | 3 | 1732|  1440| 1412 |   &#124; 839 |    740 |  732
> | 4 | 1701|  1352| 1315 |   &#124; 797 |    692 |  677
> | 5 | 1587|  1225| 1205 |   &#124; 759 |    680 |  660
> | 6 | 1414|  1157| 1121 |   &#124; 699 |     629 | 615
> | 7 | 1322|  1092| 1072 |    &#124; 667 |     604 | 594
>
> *p-values* in the table below show that our proposed 48-dimensional vector significantly improves the performance of the selection strategy over the SOAP descriptor.
>
> ||*mu*|*lumo*|
> |--| ---| --- |
> |*p-value*| 3.24 x 10E-6| 2.29 x 10E-5|
>
> It can be clearly observed that our method (whether using diversity alone or both diversity and uncertainty) outperforms SOAP descriptors. This outperformance may be attributed to the local nature of SOAP descriptors; aggregation to obtain a global descriptor can be limiting, as it may not fully capture the intricate geometric variations. Meanwhile, another important aspect to consider is permutation invariance. Through aggregation, we can ensure such symmetry. However, it is challenging and potentially a future work to determine how to effectively design a global descriptor based on SOAP that fulfills permutation invariance. We will include the results and discussion of both BatchBALD and SOAP in our paper.

---

> ### Author Response · Authors · 2024-08-10
> **Reply to Reviewer UZqH (Part II)**
>
> > $O(n^2)$ might become problematic for larger systems in case the cost starts to dominate the cost of the backbone.
>
> We refer to the case in which we assume an unbounded cutoff distance and a body-order of 2 (which is the most basic case, such as SchNet[2] that only uses relative distances), the complexity of GNN is $O(n^2)$. In general, $d< n$ with a practical cut-off distance and the model complexity is $O(d^{k-1} n)$, where $k$ is the body order. Asymptotically, we agree that our method might dominate the computational complexity of GNNs, but in real implementations, the constant term in the complexity of GNNs might be very large compared to that of our methods because of the large amount of calculations in the network, including linear and nonlinear transformations, feature aggregation and passing, etc.. Therefore, this domination will occur only with a reasonably large number of $n$. In this work, we focus on molecules, so the number of atoms ($n$) is usually small. For other scientific data like proteins, $n$ is large, but this is out of the scope of this work.
> However, we acknowledge that for large-scale datasets that contain several million conformations, the difference in complexity will increase, and we will address this in our discussion of limitations.
>
> But, most importantly, as an active learning approach, our primary focus is on minimizing the costs associated with performing annotation, rather than the computational costs tied to GNNs. These annotation costs can vary, including computational expenses like those incurred from DFT calculations ($O(n^3)$), costs from wet lab experiments, or even the time and expertise required from specialists. In many cases, we prefer to allocate more computational resources to active learning, as this investment can lead to more efficient and cost-effective labeling.
>
> > This should be noted in the limitations given that 2 molecules with similar geometries...
>
> By combining both uncertainty and diversity, we take into consideration both the chemical and geometric contexts. If we only consider the 48-dimensional vector, then, as you mentioned, the scenario you described would be a potential limitation. We acknowledge this concern and will include it in the limitations section.
>
> [1] Hyperactive learning for data-driven interatomic potentials. npj Comput Mater
>
> [2] Quantum-chemical insights from deep tensor neural networks. Nature Communications

---

> > ### Author Response · Authors · 2024-08-11
> > **Reply to Reviewer UZqH (Part III)**
> >
> > In Part I of the reply, we present the results of using SOAP descriptors to compute the diversity. We managed to conduct additional experiments using SOAP descriptors to compute the diversity matrix and supplemented it with the uncertainty component for a complete comparison with our method.
> >
> >
> > We herein present the full results. The reported values are Mean Absolute Error(MAE).
> >
> > *Abbreviations:* D= Diversity Only, B= Uncertainty + Diversity
> >
> > | Iteration |  |*mu* | |  |  |*lumo*| | |
> > |----------|------|-|---|---|----|----|--|--|
> > |   | SOAP D| Our D |SOAP B| Our B |&#124;   SOAP D| Our D |SOAP B | Our B |
> > | 1 | 2154|  1769|2057| 1741 |   &#124;   1016 |    890 | 1013 | 876
> > | 2 | 1901|  1576|1877| 1550 |   &#124; 921 |    805  |902|  799
> > | 3 | 1732|  1440|1721 |1412 |   &#124; 839 |    740 | 838|  732
> > | 4 | 1701|  1352|1539| 1315 |   &#124; 797 |    692 |791 |  677
> > | 5 | 1587|  1225|1456| 1205 |   &#124; 759 |    680 |744|   660
> > | 6 | 1414|  1157|1345| 1121 |   &#124; 699 |     629 |711|  615
> > | 7 | 1322|  1092|1280| 1072 |    &#124; 667 |     604 |681|  594
> >
> > The *p-values* in the table below further demonstrate that our proposed method (using both diversity and uncertainty) significantly improves the performance of the selection strategy compared to the SOAP descriptor (using both SOAP diversity and uncertainty).
> >
> > ||*mu*|*lumo*|
> > |--| ---| --- |
> > |*p-value*| 4.20 x 10E-6| 2.51 x 10E-6|
> >
> > It is evident that our method (using both diversity and uncertainty) outperforms the SOAP descriptors (using both SOAP diversity and uncertainty). As mentioned earlier, global SOAP descriptors may have limited capabilities in capturing essential global geometric information, leading to suboptimal performance. We will include the full results of both BatchBALD and SOAP in our paper.

---

> > > ### Comment · Reviewer_UZqH · 2024-08-12
> > >
> > > The additional results with regards to SOAP descriptors are much appreciated and support the value of the proposed diversity-based descriptor.
> > >
> > > Tuning down these bold statements and discussing the contribution properly in the context of the existing literature (as seems to be the case in light of the recent answers) further improves the paper.
> > >
> > > In light of this, I've increased my score.

---

> > > > ### Author Response · Authors · 2024-08-12
> > > >
> > > > Dear Reviewer UZqH,
> > > >
> > > > Thanks a lot for acknowledging our clarifications/experiments and raising the score.
> > > > Your comments have been constructive and helpful in improving our work. We will integrate our discussions and the new results in the next version of our paper.
> > > >
> > > > Sincerely,
> > > >
> > > > Authors,

---

### Official Review · Reviewer_yNCZ · 2024-07-15

**Soundness:** 2
**Presentation:** 3
**Contribution:** 2
**Rating:** 4
**Confidence:** 4

**Summary:**

This paper describes a way of sampling 3D graphs for active learning on
molecules, leveraging isometries.

**Strengths:**

I think this work is interesting, and potentially useful.
Clearly some kind of active learning would have useful use cases.
Also, the use of isometries could also be relevant, depending on the exact
nature of the sampling.

**Weaknesses:**

I think the main weakness of this work is that it needs stronger connections
to the actual chemistry.

* The actual elements are important. Two molecules can have similar
geometries, yet very different electronic interactions that give rise to very
different properties.

* The paper would benefit from explaining the relationship between the
different isometries and actual molecules and chemical properties.

* The space of molecules is very large.  The tests hold back part of known
datasets as unlabeled molecules.  But the most useful new labels may not be
in the QM9 dataset, but may be in some other dataset, or even a novel
molecule.  Thus, a more interesting algorithm would be one that identified a
molecule X as as important new molecule to add, allowing some search for it.
Of course, this would raise a number of other important issues, such as
generating the new molecule X, etc.

* The paper does not test against 2-D methods.  Not sure why.  Perhaps they
perform well.

**Questions:**

* How does sampling the graphs relate to sampling the interactions that are
responsible for properties?

* Is chirality considered?

* Why do the graphs in Figure 3 not start at the same loss?  Before any active
learning has occured, shouldn't that the situation?

* Are the other AL techniques also designed to be used in this iterative manner?
If not, I think a better baseline test would be to use the same number of
samples.

**Limitations:**

The paper does not adequately address limitations. I would imagine one would be that it does not consider the actual elements of each atom. In other words, two molecules with similar geometries are considered similar, even though they could be very different chemically.

What about periodic structures?

Also, what if there is not a pre-existing source of valid, but unlabeled molecules? A common situation is that there are only a small number of molecules known at all.

---

> ### Author Rebuttal · Authors · 2024-08-01
>
> >W1: about actual atoms
>
> Our framework contains two components for selecting important molecules: diversity and uncertainty. **We do consider the actual atoms in the uncertainty part, as the atom types are embedded into node features.**
>
> Our diversity component focuses on molecular geometries, **which essentially reflect the interactions among atoms in a molecule.**
> Thus, 3D shapes significantly influence molecular properties [1, 2]. As a concrete example, when predicting the important HOMO-LUMO gap of QM9 using MAE (lower is better), using data without 3D geometries produces *1.23-3.28* [3], while using data with 3D info significantly improves it to *0.03-0.06* [4]. The GWL test also focuses on geometries, and 3D GNNs aim to integrate geometric information into the learning process.
>
> **In summary, by combining both uncertainty and diversity, we take both the chemical contexts and geometric contexts into consideration.** We will emphasize this in the next version.
>
> > W2: the relationship between different isometries, actual molecules, and chemical properties
>
> **Full geometric information is important for computing chemical properties.** As we discussed in lines 172-179, different isometries are mainly related to the completeness of the geometric information and the universal approximation of 3D GNNs. For example, simply using the distances (Reference Distance Isometry) cannot recognize two molecules with the same edge lengths but different angles, which is expected to produce suboptimal property predictions. Using angle information (Triangular Isometry), for example, improves the MAE of the HOMO-LUMO gap from *0.063 to 0.033* [4].
>
> >W3: extend to different datasets and even generate new molecules
>
> Very insightful view!
>
> First, we can definitely extend QM9 to new datasets like MD17, as our active selection is applicable to all 3D molecules. We do it separately because we follow the fashion of the community and it's easy for evaluation.
>
> Second, this study falls into molecular property prediction, and molecular generation is a different topic. We agree that extending our algorithm to identify novel molecules can be highly beneficial. **As you mentioned, this approach could raise other issues, such as the validation of the generated molecules, which is far beyond the scope of this work.** Therefore, we kindly request not to view this as a weakness of our work.
>
> >W4: does not test against 2-D method
>
> The scope of this work is 3D molecular learning based on 3D GNNs. As we have pointed out previously, the 3D conformations of molecules determine their properties; thus, **3D GNNs outperform 2D GNNs by a large margin**. Formulating molecules as 3D graphs introduces new challenges for active selection criteria, which motivates our current work.
>
>
> > Q1: sampling the graphs and sampling the interactions
>
> Interactions of atoms are reflected by the 3D positions of all atoms in a graph, which is exactly the reason why we formulate molecules as 3D graphs. We design strategies to select the most informative molecules for efficient learning given a limited budget, this is, we select molecules with the most salient interatomic features.
>
> > Q2: chirality
>
> No. Our isometries are defined in the E(3) group instead of the SE(3) group, and the only difference is reflection, i.e., chirality in chemistry. First, AL selects more informative samples, and we want chiral molecules to have similar informativeness (diversity scores). Chirality can significantly affect molecular properties. If we treat a molecule as an important sample to learn from but don't treat its chiral counterparts as important, the chiral molecules are less likely to be seen in training. This can result in a GNN that is very biased due to the lack of diverse chiral data during training.
>
> Moreover, it is for efficiency. Some recent GNNs (like GemNet [5]) can recognize chiral molecules, but **the complexity is $O(n^3)$ while ours is $O(n^2)$.**
>
> > Q3: Figure 3 not start at the same loss
>
> Since all the methods start with the same initial labeled training set, their starting MAE values on the test set will be the same. **We have therefore plotted the MAE values from the first iteration onwards, to focus on the comparative performance of the methods after they start selecting samples using AL.**
>
> > Q4: Iterative manner of AL
>
> **Active Learning (AL) techniques are designed to operate in an iterative manner (see mainstream AL papers mentioned in the Related Work section).** A budget $k$ is imposed on the number of unlabeled samples that can be queried for annotation in each iteration.**We followed this conventional setup**.
>
>
>
> > L1: actual elements of each atom
>
> Please refer to W1; **our method consists of two parts that take into account both geometric and atomic information**. Diversity in this paper focuses on geometries. Actually, we did conduct experiments for diversity that, **a vector for atom types** was concatenated to the current geometric vector, but the performance remained the same. This is mainly because, considering chemical priors like interatomic forces, it's nearly impossible for two different stable molecules to have the same 3D shape.
>
> > L2: periodic structures
>
> **Periodic structures are only for crystal materials** (see famous methods CGCNN, MEGNET, ALIGNN etc). Our focus is on small molecules, so it's beyond our scope of interest. We will discuss this.
>
> > L3: unknown molecules
>
> This can be a very interesting perspective and a new research direction that is outside the scope of this work. We will make sure to include this in our limitations in the paper.
>
> Reference
>
> [1] G. T., et al. Principles governing amino acid composition. Journal of Molecular Biology
>
> [2] B. J., et al. Torsional Diffusion for Molecular Conformer Generation. NeurIPS'22
>
> [3] Y. L., et al. Spherical message passing for 3D molecular graphs. ICLR'22
>
> [4] J. G., et al. Neural message passing for quantum chemistry. ICML'17
>
> [5] J. G., et al. GemNet. NeurIPS'21

---

> > ### Comment · Reviewer_yNCZ · 2024-08-14
> > **Completely agree that 3D is important**
> >
> > Thank you for the rebuttal. I completely agree that 3D geometry is crucial. But it is not clear to me why 2 molecular graphs with different elements is not considered diverse, but rather uncertain. To a chemist, those are two completely different molecules, even though the graph, with elements removed, is the same. In other words, the formalism for graphs, as far as I can tell, described on line 227, does not include the elements.
> >
> > After careful thought, I think this work has promise, but will stay with my rating.

---

> ### Author Response · Authors · 2024-08-13
> **The discussion period is closing in 32 hours**
>
> Dear Reviewers yNCZ,
>
> Thank you for your insightful comments, which have been very helpful in improving the clarity of our work. As the **reviewer-author discussion stage is ending in 32 hours**, we kindly remind you that we have provided a detailed response to address your concerns. We hope our clarifications have resolved the issues to your satisfaction.
>
> If you believe your concerns have been adequately addressed, we kindly ask you to reconsider your scores. However, if there are any aspects that still need further clarification, please let us know, and we are more than willing to discuss them in detail.
>
> Sincerely,
>
> Authors

---

> ### Author Response · Authors · 2024-08-13
> **We are eagerly awaiting your response**
>
> Dear Reviewer yNCZ,
>
> Thanks again for your insightful comments, which we believe will improve the clarity of our work. Regarding your concerns, we've made efforts to address them in detail in our rebuttal. As you are the only reviewer who has not responded to our initial rebuttal, we sincerely hope you can check it at your earliest convenience.
>
> Since the discussion period is approaching its end, we hope you can let us know if we have addressed your critical points and reconsider the score if we have. Meanwhile, we welcome any additional questions and wish to discuss them in detail.
>
> Sincerely,
>
> Authors

---

> ### Author Response · Authors · 2024-08-14
> **Follow up response to Reviewer yNCZ**
>
> Thanks for your comment and discussion! While we highly respect your thought to maintain the rating, we still hope you can reconsider if other aspects in your initial set of comments have been clarified and addressed.
>
> ----------------------------------
> Regarding this new comment, we provide clarifications below:
>
>
> 1) **We do consider elements in $\mathbf{G}$ in line 227.** In $\mathbf{G}=(V, E, P)$, $V$ denotes the set of elements (atom types), **as mentioned in the same line**. Particularly, this is a widely adopted setting in mainstream 3D GNN models for molecular learning [1,2,3,4].
>
> 2) A typical active learning process includes two stages: the selection stage (selecting the most informative molecules to annotate, to reduce the annotation budget), and representation learning (like molecular property prediction and molecular dynamic simulation). **Elements are considered in both stages.**
>
> 3) The selection stage (Eq. (4) in the paper, lines 266-272) combines two strategies (diversity and uncertainty) to select the most informative molecules. **Elements are considered in the uncertainty part.** Following classical geometric descriptors, diversity focuses on geometric descriptors. **Hence, the overall selection stage (Eq. (4), where *r* denotes uncertainty scores for molecules) considers elements.**
>
> 4) For the example you gave - two molecules with the same graph but different elements, $\mathbf{G}$ in line 227 would be different, thus uncertainty would be very different, through Eq. (4)  the informativeness of these two molecules are totally different.
>
>
> 5) Last but not least, for our diversity descriptor, we included a comparison with a well-known geometric descriptor in chemistry, the SOAP descriptor [5, 6, 7], which produces descriptors that describe local atomic environments using spherical harmonics and radial basis functions. **SOAP considers both geometric information and elements (species).** Our results reveal that our diversity component outperforms SOAP; our overall method (using both diversity and uncertainty as in Eq. (4)) also outperforms the SOAP descriptors (using both SOAP diversity and uncertainty). This outperformance can be attributed to the local nature of SOAP descriptors. Our work is among the first to consider a global 3D geometric descriptor for molecular learning, combined with an uncertainty component that considers chemical contexts (elements etc), enabling more accurate and robust quantification of the informativeness of an unseen (by the GNN) molecule.
>
>
> &nbsp;&nbsp;&nbsp;&nbsp;&nbsp;&nbsp; We herein present the results. The reported values are Mean Absolute Error (MAE, lower is better).
>
> &nbsp;&nbsp;&nbsp;&nbsp;&nbsp;&nbsp; *Abbreviations:* D= Diversity Only, B= Uncertainty + Diversity
>
> | Iteration |  |*mu* | |  |  |*lumo*| | |
> |----------|------|-|---|---|----|----|--|--|
> |   | SOAP D| Our D |SOAP B| Our B |&#124;   SOAP D| Our D |SOAP B | Our B |
> | 1 | 2154|  1769|2057| 1741 |   &#124;   1016 |    890 | 1013 | 876
> | 2 | 1901|  1576|1877| 1550 |   &#124; 921 |    805  |902|  799
> | 3 | 1732|  1440|1721 |1412 |   &#124; 839 |    740 | 838|  732
> | 4 | 1701|  1352|1539| 1315 |   &#124; 797 |    692 |791 |  677
> | 5 | 1587|  1225|1456| 1205 |   &#124; 759 |    680 |744|   660
> | 6 | 1414|  1157|1345| 1121 |   &#124; 699 |     629 |711|  615
> | 7 | 1322|  1092|1280| 1072 |    &#124; 667 |     604 |681|  594
>
> &nbsp;&nbsp;&nbsp;&nbsp;&nbsp;&nbsp; The *p-values* in the table below further demonstrate that our proposed method (using both diversity and uncertainty) significantly improves the performance of the selection strategy compared to the SOAP descriptor (using both SOAP diversity and uncertainty).
>
> ||*mu*|*lumo*|
> |--| ---| --- |
> |*p-value*| 4.20 x 10E-6| 2.51 x 10E-6|
>
> [1] K.S., et al. SchNet: A continuous-filter convolutional neural network for modeling quantum interactions. NIPS'17
>
> [2] J. G., et al. Neural message passing for quantum chemistry. ICML'17
>
> [3] Y. L., et al. Spherical message passing for 3D molecular graphs. ICLR'22
>
> [4]  J. G., et al. GemNet: Universal Directional Graph Neural Networks for Molecules. NeurIPS'21
>
> [5] S.D., et al. Comparing molecules and solids across structural and alchemical space. Physical Chemistry Chemical Physics
>
> [6] M.J., et al.Machine learning hydrogen adsorption on nanoclusters through structural descriptors. npj Computational Materials
>
> [7] A. B., el al. On representing chemical environments. Physical Review B - Condensed Matter and Materials Physics
>
> -----------------
> Hope these clarifications can address your concern. If you have further questions, don't hesitate to let us know, and we are always ready to discuss them in detail.

---

### Author Rebuttal · Authors · 2024-08-07

We thank the reviewers for their invaluable comments and suggestions. In this global response, we would like to clarify a few points and present new results based on the feedback received.

### Clarification
To start our rebuttal, we would like to clarify a few points about our method.
> Our model considers both geometry and chemical contexts to achieve the best performance

Firstly, our method consists of two parts: uncertainty and diversity. The diversity component, based on our proposed geometric isometries, aims to focus on diverse molecules, thereby capturing a wide range of chemical properties. The diversity part is our major contribution. As we will discuss later, the results of studying the individual contributions of uncertainty and diversity to performance show that the diversity component is highly effective.

Additionally, our method includes an uncertainty component, which aims to quantify and incorporate the uncertainty in our model's predictions. The uncertainty part also takes chemical contexts (like atom types) into account as node features, enhancing the model's ability to recognize and better learn uncertain chemical interactions. **Therefore, our method considers both geometry and chemical contexts to achieve the best performance.**

### Additional Results
To this end, we would like to discuss some additional results we obtained, which can be found in the **attached PDF file**. References will be made to this attached file rather than the paper unless otherwise specified.

> 1. Comparison with a new baseline, BatchBALD (as suggested by Reviewer UZqH)

In Figure 1, we included the results for a new baseline, BatchBALD[1], adapted for our regression setting following [2]. Similar to the uncertainty component in our method, BatchBALD is based on Bayesian uncertainty, and it serves as an important baseline in this venue. As observed in the results, **our method consistently achieves a lower MAE (the lower the better) at any given AL iteration compared to BatchBALD**. For our analysis of other baselines, please refer to the paper. In addition, as a complement to the result above, we updated Table 1 with p-values obtained using paired t-tests between our method and all baselines, now including BatchBALD. **The performance improvement achieved by our method is statistically significant** ($p << 0.001$) for all $4$ properties tested against BatchBALD.

Given that BatchBALD is also a Bayesian-based uncertainty approach, this comparison further highlights the robustness and effectiveness of our method, especially the combination of uncertainty and diversity. To this end, it is important to emphasize that our method has two components: uncertainty and diversity. The diversity component, which is our main contribution, plays a more fundamental role in improving performance. This is because capturing 3D atomic geometries is crucial for accurately modeling and understanding molecular interactions. By incorporating diversity, our method ensures a more comprehensive selection of informative samples, leading to better overall results.

> 2. On the individual impact of diversity and uncertainty components (as suggested by Reviewer Abqb and Reviewer JSdU)

In Figure 2, we present a study on the individual impact of the diversity and uncertainty components. It is clear that our proposed method outperforms the individual use of diversity or uncertainty alone. The key to this outperformance lies in our method’s dual focus on both geometric importance and chemical contexts. Moreover, it can be observed that the diversity component alone shows strong performance, it is only slightly less effective than our method because it **captures the geometries of molecules, which are fundamental in distinguishing different molecules with different properties**. On top of this, we also conducted statistical tests to conclude that **the improvement of our method is significant** compared to only diversity or only uncertainty ($p << 0.001$) in Table 2.

> 3. Vectorized implementation of our methods and updated timing results (in response to the concerns of Reviewer UZqH and Reviewer Abqb)

In Table 3 of the PDF attached, we provided updated results on the average time taken by our method. Compared with the original results in the paper (see *Table 2* in the paper), the computational time is reduced from 15 minutes slower than random to 11 minutes slower than random. This improvement in calculation time is due to **vectorizing our implementation and utilizing the GPU to perform calculations for diversity matrix computation**. Vectorization accelerates computations by processing multiple data points parallelly, a task at which GPUs excel. For instance, deep neural networks also benefit from parallelization, achieving faster performance on GPUs compared to CPUs. A simple example is calculating the inner product of two vectors. Without vectorization, we compute the products of corresponding elements one by one and then sum them sequentially. With vectorization, the GPU performs element-wise multiplications in parallel and then computes the sum in a single, efficient reduction step.

It can be seen that our sampling approach is highly effective and efficient. Note that the Coreset approach is an important diversity-based baseline, and our sampling approach is much more efficient than it (127 vs 64.9 minutes). The Random approach does not involve any sophisticated sampling strategy, and our method is just slightly more expensive than it (53 vs 64.9 minutes). Thus, we believe our sampling approach is readily applicable to larger datasets.


[1] Andreas Kirsch, et al. BatchBALD: Efficient and Diverse Batch Acquisition for Deep Bayesian Active Learning, NIPS

[2] D.H.,et al. A framework and benchmark for deep batch active learning for regression, JMLR

---

### Author Response · Authors · 2024-08-12
**We are looking forward to your response**

Dear Reviewers,

Thank you for your insightful and constructive comments, which have been very helpful in improving our work. As the reviewer-author discussion stage is ending soon, we kindly remind you that we have provided a detailed response to address your concerns (both individual response and global response). We hope our clarifications, along with the additional experimental results and efforts we've included, have resolved the issues to your satisfaction.

If you believe your concerns have been adequately addressed, we kindly ask you to reconsider your scores. However, if there are any aspects that still need further clarification, please let us know, and we are more than willing to discuss them in detail.

Sincerely,

Authors

---

### Decision · Program_Chairs · 2024-09-25

**Decision:**

Accept (poster)

**Comment:**

This paper presents an active learning method for 3D molecular graphs, where the diversity of molecules are measured according to 3D graph isomorphism analysis. Experimental results on two data sets prove the effectiveness of the proposed approach.

In general, the reviewers like the novelty of the proposed approach. There are a few aspects that the authors need to address before publication: (1) the claims are too strong; (2) more baselines should be included, e.g., 2D-graph based approaches.